

# Revisiting Wess-Zumino-Witten terms

**Yasunori Lee[1], Kantaro Ohmori[2] and Yuji Tachikawa[1]**

**1** Kavli Institute for the Physics and Mathematics of the Universe (WPI),
University of Tokyo, Kashiwa, Chiba 277-8583, Japan
**2** Simons Center for Geometry and Physics, SUNY, Stony Brook, NY 11794, USA

## Abstract

We revisit various topological issues concerning four-dimensional ungauged and gauged Wess-Zumino-Witten (WZW) terms for $SU$ and $SO$ quantum chromodynamics (QCD), from the modern bordism point of view. We explain, for example, why the definition of the 4$d$ WZW terms requires the spin structure. We also discuss how the mixed anomaly involving the 1-form symmetry of $SO$ QCD is reproduced in the low-energy sigma model.

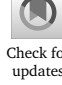

# 1 Introduction and summary

Massless quantum chromodynamics (QCD), when the number of flavors $N_f$ is sufficiently smaller than the number of colors $N_c$, is described in the infrared in part by a non-linear sigma model parameterizing the Nambu-Goldstone bosons associated to the spontaneous symmetry breaking of the flavor symmetry. The anomaly of the flavor symmetry is non-zero, and therefore from 't Hooft's anomaly matching [1], it also needs to be realized in the low-energy sigma model. This is provided by a certain term defined on the sigma model which was originally identified by Wess and Zumino [2],[1] whose topological significance was later brought to the fore by Witten in [3]. This term is still non-trivial after turning off the background gauge field for the flavor symmetry as noted by [3], and the terms of this general form were earlier described independently by Novikov in [4]. The aim of this paper is to revisit these topological

---

[1]Note that this was before 't Hooft formulated his anomaly matching.

terms, which we call ungauged and gauged Wess-Zumino-Witten (WZW) terms,[2] for $SU$ and $SO$ QCD.[3]

The first concrete issue we would like to address is the following. In the standard references which introduce the WZW terms, the quantization of the overall coefficient is determined by considering the sigma model configurations on a flat space or a sphere. It is therefore not clear whether the WZW terms are well-defined on an arbitrary spacetime with an arbitrary sigma model configuration. We will show that the consistency of the ungauged WZW terms requires that a spacetime is equipped with a spin structure, and will examine possible additional complications present in the gauged WZW terms.

We carry out our re-analysis in the context of the recent improved understanding of invertible phases, which are quantum field theories depending on some set of background fields such that the partition function is always a complex number of absolute value 1. Invertible topological phases depending on background gauge fields are in essence equivalent to symmetry-protected topological (SPT) phases, which were originally introduced in the condensed matter literature, and have been extensively studied there. In contrast, WZW terms are examples of invertible phases depending on scalar background fields, i.e. maps from the spacetime to the sigma model target space. Such invertible phases and the corresponding anomalies have only recently begun to be studied in the literature, see e.g. [5–11]. That said, the general classification of invertible phases established in the last several years [12–16] is equally applicable in both cases, and will facilitate our analysis.

We note here that the importance of regarding the topological terms in the effective action as a bona fide QFT whose partition function is always invertible was first stressed in a paper by Freed and Moore [17], where the phrase *invertible quantum field theories* was first introduced. We also note that the analysis of the WZW term of the $SU$ QCD from this point of view was already given in [18] with more mathematical rigor. Unfortunately, that paper was written long before the general understanding of invertible phase was achieved, and therefore was not very easy to read. We hope that our discussion here provides a more readable introduction to this general framework. [4]

We will also discuss topological solitons in the low-energy sigma models. Recall that baryons in $SU$ QCD are solitonic particles in the low-energy sigma model [19, 20], whereas $\mathbb{Z}_2$-valued electric flux tubes in $Spin$ QCD are solitonic strings [20]. The $U(1)$ baryonic symmetry of $SU$ QCD has a mixed anomaly with the flavor symmetry. This mixed anomaly of $SU$ QCD is also represented in the low-energy sigma model [21–23]. We would like to describe its analogue in the case of $\mathbb{Z}_2$-valued flux tubes of $SO$ QCD.

For this we need the concept of $p$-form symmetries: while point-like operators are charged under *ordinary* symmetries, higher-dimensional operators are charged under *higher-form* symmetries. Denoting the dimensionality of charged operators by $p$, these two are uniformly treated as $p$-form symmetries [24]. In this language, solitonic particles are charged under the $U(1)$ 0-form symmetry of $SU$ QCD, and solitonic strings are charged under the $\mathbb{Z}_2$ 1-form symmetry of $SO$ QCD. Then what we need to do is understand the mixed anomaly of the $\mathbb{Z}_2$ 1-form symmetry with other symmetries in the ultraviolet, and describe how it is represented

---

[2]Novikov [4] only discussed the ungauged version while Wess and Zumino [2] only discussed the gauged version. Therefore it might be more logical to call the ungauged WZW term as the Novikov-Witten term and the gauged WZW term simply as the WZW term. Here we follow the established convention in the literature.

[3]We will use the phrase *G QCD* as a generic name for QCD whose gauge algebra is the Lie algebra of $G$. Some of the discussions in our paper do depend on the choice of the gauge group and the discrete theta angle, not just on the algebra. In those occasions, we explicitly specify our choice, such as *Spin QCD*.

[4]In particular, the reference [18] used a generalized cohomology theory which was simply called $E^\bullet$ in that paper. It is a truncation of the cohomology theory $(D\Omega_{\mathrm{spin}})^\bullet$, which is the Anderson dual of the spin bordism group and is now understood to be the correct cohomology theory classifying the spin invertible phases. From the perspective we have now, the appearance of $E^\bullet$ can better be understood by first considering the use of $(D\Omega_{\mathrm{spin}})^\bullet$, which can be physically motivated as we do in Sec. 2.

in the low-energy sigma model. The first task was very recently performed in [25] which we briefly review.[5] Our remaining task is then to demonstrate how it is realized in the infrared. This requires us to take into account the $\mathbb{Z}_2$ 1-form gauge theory present in the low-energy limit of the $SO$ QCD.

The rest of the paper is organized as follows. We start in Sec. 2 by studying the ungauged WZW terms from a modern perspective. After introducing the WZW terms at the level of cohomology, we will see that the WZW terms for QCD are more sophisticated and require the spin structure for their definitions. This section can be read as a gentle introduction to the modern understanding of invertible phases and their relationship with the Anderson dual of bordism groups. In Sec. 3, we review the anomaly of four dimensional $SU$ and $SO$ QCD. In particular, we review the fact that $SO(2n_c)$ QCD has 1-form symmetry in addition to other ordinary 0-form symmetries, and that there is a mixed 't Hooft anomaly between them, as recently pointed out in [25]. In Sec. 4, we study possible topological issues in the definition of gauged WZW terms. After developing a general framework to analyze such issues, we will see that the gauged WZW terms for $SU$ and $SO$ QCD are well-defined almost automatically in the presence of spin structure. In Sec. 5, we include the solitonic objects of the low-energy sigma models in our analysis. For $SU$ QCD, they are skyrmions associated to $\pi_3$ of the sigma model target space, and for $Spin$ QCD, they are electric flux tubes associated to $\pi_2$. We will see how the mixed anomalies associated to their conserved charges reproduce those in the ultraviolet. Finally, we have four appendices. In Appendix A we collect basic information on cohomology groups of relevant homogeneous spaces and classifying spaces; in Appendices B and C we compute required bordism groups via Atiyah-Hirzebruch spectral sequence and Adams spectral sequence respectively; and in Appendix D we discuss subtler issues concerning $SO$ WZW terms.

## 2  Ungauged WZW terms

Let us first discuss ungauged WZW terms from a modern perspective. In particular, we would like to explain why one needs (co)bordisms, rather than (co)homologies, to describe these terms.

We start in Sec. 2.1 by viewing $U(1)$ gauge fields as $1d$ WZW terms. In Sec. 2.2 we will turn to $B$-fields and $2d$ WZW terms. Then in Sec. 2.3 these two basic cases are abstracted to cohomological WZW terms in general dimensions. In Sec. 2.4, we begin our study of $4d$ WZW terms, by taking up the WZW terms for $SU$ QCD with $N_f \geq 3$. We will see that the spin structure is necessary to ensure that the integral of a cohomology class is always even. We then analyze in Sec. 2.5 the $SU$ WZW terms for $N_f = 2$, whose form looks rather different from the WZW terms for $N_f \geq 3$. We will discuss how they are related. In Sec. 2.6 we move on to discuss the WZW terms for $SO$ QCD for generic $N_f$. This time the spin structure is necessary to ensure that the integral of a cohomology class is a multiple of four. We also point out the existence of a subtle torsion part. The next section Sec. 2.7 is devoted to the discussion of the $SO$ WZW term for $N_f = 2$, which again shows special features. The final subsection, Sec. 2.8, combines all the preceding analyses into a general prescription, which describes and classifies invertible phases depending on a target manifold $X$.

---

[5]A closely related 1-form symmetry anomaly in pure $SU(rs)/\mathbb{Z}_s$ Yang-Mills theory was studied in [26].

## 2.1 $U(1)$ connections

We will start with the simplest example, as was also done in [3]. Consider a $(0+1)d$ theory with a scalar field $\phi$ taking values in a manifold $X$. This is simply a convoluted way of referring to a quantum mechanical particle moving on $X$. Let us make the particle electrically charged by coupling to a $U(1)$ gauge connection $A$ on $X$. We denote the worldline of the particle by a scalar field $\phi : S^1 \to X$. Then the contribution $e^{-S[\phi,A]}$ of the connection to the exponentiated action is given by the holonomy along $S^1$ of the pull-back $\phi^*(A)$ of the gauge connection. Physicists often write this somewhat imprecisely as

$$e^{-S[\phi,A]} = e^{i \int_{S^1} \phi^*(A)} = e^{i \int_{\phi(S^1)} A}, \tag{2.1}$$

as if $A$ were always a globally well-defined one-form.

Let us interpret the holonomy in a way useful for our generalizations later. First, suppose that the loop $\phi_0 : S^1 \to X$ is contractible within $X$, or equivalently, that it can be extended to $\phi : D^2 \to X$, where $D^2$ is a two-dimensional disk. We can then write the holonomy as

$$e^{-S[\phi_0,A]} = e^{i \int_{\phi(D^2)} F}, \tag{2.2}$$

where $F$ is the field strength, or equivalently the curvature of the $U(1)$ connection $A$. The right hand side does not depend on the choice of the extension $\phi$, as can be shown as follows. Take another extension $\phi' : D^2 \to X$. Then we have

$$e^{i \int_{\phi(D^2)} F} / e^{i \int_{\phi'(D^2)} F} = e^{i \int_Y F}. \tag{2.3}$$

Here, we let $Y = \phi(D^2) \cup \overline{\phi'(D^2)}$, where $\overline{M}$ is the orientation reversal of $M$ and the union is taken by identifying their boundaries. This makes $Y$ an image of a sphere in $X$. Then we have $\int_Y F = 2\pi n$ for some integer $n$, showing that the right hand side of (2.3) is indeed 1.

We repeat the salient points above:

- We extend the spacetime $S^1$ and the scalar field $\phi : S^1 \to X$ to an auxiliary higher-dimensional space $D^2$ and the scalar field $\phi : D^2 \to X$, which are then used to define the coupling (2.2).

- We can then show that the expression (2.2) does not depend on the extension, thanks to the fact that the integral of $F$ over closed manifolds is $2\pi$ times integers.

These are the two basic features of the WZW coupling.

The same technique allows us to compare the holonomy along two configurations $\phi_0 : S^1 \to X$ and $\phi_1 : S^1 \to X$ deformable to each other, in the sense that there is a map $\phi : S^1 \times [0,1] \to X$ such that the boundary values are equal to $\phi_{0,1}$, respectively. Then we have

$$e^{-S[\phi_0,A]} / e^{-S[\phi_1,A]} = e^{i \int_{S^1 \times [0,1]} \phi^*(F)} = e^{i \int_{\phi(S^1 \times [0,1])} F}. \tag{2.4}$$

The right hand side is independent of the choice of $\phi$ interpolating $\phi_0$ and $\phi_1$. Indeed, given another $\phi' : S^1 \times [0,1] \to X$ interpolating $\phi_{0,1}$, we can consider

$$e^{i \int_{\phi(S^1 \times [0,1])} F} / e^{i \int_{\phi'(S^1 \times [0,1])} F} = e^{i \int_Y F}, \tag{2.5}$$

where $Y = \phi(S^1 \times [0,1]) \cup \overline{\phi'(S^1 \times [0,1])}$ is a 2-cycle in $X$. Again, $\int_Y F = 2\pi n$ guarantees that this is indeed 1.

Next, suppose that the field strength $F$ vanishes. Then the holonomy does not depend on deformations of the loop $\phi(S^1) \subset X$. Stated differently, the holonomy determines a character $\chi_A : H_1(X; \mathbb{Z}) \to U(1)$, and as a result we have

$$e^{-S[\phi,A]} = \chi_A\big([\phi(S^1)]\big). \tag{2.6}$$

A general $U(1)$ connection can roughly be regarded as a certain combination of two extremes (2.2) and (2.6).

## 2.2 $2d$ WZW terms

Let us next consider a $(1+1)d$ QFT describing a string moving within a manifold. We denote the worldsheet of the string by $M_2$, the target manifold by $X$, and the embedding by $\phi : M_2 \to X$. An important ingredient of string theory is the $B$-field, whose contribution to the exponentiated action is its holonomy $e^{-S[\phi(M_2),B]}$. This can be written as

$$e^{-S[\phi(M_2),B]} = e^{i \int_{\phi(M_2)} B}, \tag{2.7}$$

when $B$ is a globally well-defined two-form.

More generally, a $B$-field has a field strength $H$ which is a closed 3-form, such that we have

$$\int_{[Y]} H \in 2\pi\mathbb{Z} \tag{2.8}$$

for any 3-cycle $[Y] \in H_3(X;\mathbb{Z})$. Now, suppose $\phi : M_2 \to X$ and $\phi' : M_2' \to X$ are *bordant*, i.e. there exists a $W_3$ such that $\partial W_3 = M_2 \sqcup \overline{M_2'}$ so that there is a $\phi : W_3 \to X$ whose restrictions on the boundaries give $\phi$ and $\phi'$ respectively. Then we have

$$e^{-S[\phi(M_2),B]}/e^{-S[\phi'(M_2'),B]} = e^{i \int_{\phi(W_3)} H}. \tag{2.9}$$

The right hand side does not depend on the choice of the extension, thanks to the condition (2.8).

In particular, when $\phi : M_2 \to X$ is contractible and extensible to $\phi : W_3 \to X$ such that $\partial W_3 = M_2$, the property above suffices to determine the holonomy, and indeed we have

$$e^{-S[\phi(M_2),B]} = e^{i \int_{\phi(W_3)} H}. \tag{2.10}$$

For the $2d$ WZW model, $X$ is a group manifold of a compact Lie group $G$. Assume $G$ is simple and simply connected. Then $H^3(X;\mathbb{R}) \simeq \mathbb{R}$, and there is a $G$-invariant 3-form $\Gamma_3$ generating it. We normalize it so that $\int_Y \Gamma_3 = 2\pi$, where $[Y]$ is a generator of $H_3(X;\mathbb{Z}) \simeq \mathbb{Z}$. Any $\phi : M_2 \to X$ is contractible in this case. Choosing an extension $\phi : D_3 \to X$, the WZW term is given by

$$e^{-S} = e^{ik \int_{D_3} \phi^*(\Gamma_3)}, \tag{2.11}$$

where the integer $k$ is called the *level*.[6]

In the other extreme, consider the case when $H$ vanishes. Then the $B$-field determines a character $\chi_B : H_2(X;\mathbb{Z}) \to U(1)$, and therefore the holonomy is given by

$$e^{-S[\phi(M_2),B]} = \chi_B\big([\phi(M_2)]\big). \tag{2.12}$$

The $U(1)$ gauge fields $A$ are mathematically formalized as $U(1)$ bundles. The $B$-fields are formalized as gerbes in mathematical literature.

## 2.3 WZW terms at the level of (co)homology

The construction above can be generalized to arbitrary spacetime dimensions as follows [4,27]. Before proceeding, we emphasize that some parts of the descriptions in this subsection will be superseded in Sec. 2.8, by replacing homology groups by bordism groups.

---

[6]This WZW term is invariant under the global $G$ transformation, but in the general case when $X$ has an isometry $G$ and when $\Gamma_3$ is invariant under $G$, it is not necessarily the case that the resulting ungauged WZW term is invariant under $G$, when the image of $M_2$ within $X$ is homologically non-trivial. For details, see [28].

Consider a $d$-dimensional theory with a scalar field $\phi$ taking values in a manifold $X$. We can now consider a $d$-form gauge field $C$ on $X$, which has the following features. First, it has an associated closed $(d+1)$-form field strength $G$, such that when the scalar field $\phi : M_d \to X$ is extensible to $\phi : W_{d+1} \to X$ with $\partial W_{d+1} = M_d$, the coupling is given by

$$e^{-S[\phi(M_d),C]} = e^{i \int_{\phi(W_{d+1})} G}.$$ (2.13)

For this coupling to be independent of the extension, we require that

$$\int_{[Y]} G \in 2\pi\mathbb{Z}$$ (2.14)

for all $[Y] \in H_{d+1}(X;\mathbb{Z})$. Second, when the field strength $G$ vanishes so that the $d$-form gauge field is flat, the coupling is given by

$$e^{-S} = \chi\big([\phi(M_d)]\big),$$ (2.15)

where

$$\chi : H_d(X;\mathbb{Z}) \to U(1)$$ (2.16)

is a character. The mathematically precise formulation of these ideas is known as *differential characters* and/or *differential cohomology*.[7] It is simply a $U(1)$ connection when $d = 1$, while it is a gerbe or a $B$-field when $d = 2$.

The topological class of a $d$-form gauge field $C$ is given by a class $c = [G/2\pi] \in H^{d+1}(X;\mathbb{Z})$; when $d = 1$ this $c$ is the *first Chern class* of the $U(1)$ gauge connection, and when $d = 2$ this $c$ is often called the *Dixmier-Douady class* of the gerbe. The important fact is that the information contained in $c$ can be decomposed to pieces corresponding to (2.13) and (2.16).

To see this, use the universal coefficient theorem of (co)homology which says that $H^{d+1}(X;\mathbb{Z})$ sits in the short exact sequence

$$0 \to \text{Ext}_{\mathbb{Z}}(H_d(X;\mathbb{Z}),\mathbb{Z}) \xrightarrow{b} H^{d+1}(X;\mathbb{Z}) \xrightarrow{a} \text{Hom}_{\mathbb{Z}}(H_{d+1}(X;\mathbb{Z}),\mathbb{Z}) \to 0,$$ (2.17)

where for a finitely generated Abelian group $A$ we have

$$\text{Ext}_{\mathbb{Z}}(A,\mathbb{Z}) = \text{Hom}(\text{Tors}\,A, U(1)), \qquad \text{Hom}_{\mathbb{Z}}(A,\mathbb{Z}) = \text{Hom}(\text{Free}\,A, \mathbb{Z}),$$ (2.18)

where $\text{Tors}\,A$ is the torsion subgroup of $A$ and $\text{Free}\,A = A/\text{Tors}\,A$ is the free part of $A$, so that when $A = \mathbb{Z}^n \oplus \mathbb{Z}_{n_1} \oplus \cdots \oplus \mathbb{Z}_{n_k}$, $\text{Tors}\,A = \mathbb{Z}_{n_1} \oplus \cdots \mathbb{Z}_{n_k}$ and $\text{Free}\,A = \mathbb{Z}^n$.[8]

Then, $a(c) : H_{d+1}(X;\mathbb{Z}) \to \mathbb{Z}$ is the mapping $\int_{[Y]} G/(2\pi)$ for $[Y] \in H_{d+1}(X;\mathbb{Z})$, corresponding to the part (2.13). When $a(c)$ vanishes, the gauge field can be continuously deformed to a flat one, whose information is captured by (2.16). The holonomy assigned to the free part of $H_d(X;\mathbb{Z})$ can be continuously deformed to a trivial one, and therefore the topological class of $c$ when $a(c) = 0$ is specified by $\text{Hom}(\text{Tors}\,H_d(X;\mathbb{Z}), U(1))$.

We further note that the exact sequence (2.17) can be identified with a part of the long exact sequence associated to the change of coefficients

$$0 \longrightarrow \mathbb{Z} \longrightarrow \mathbb{R} \xrightarrow{\pi} U(1) \longrightarrow 0.$$ (2.19)

---

[7]The concept was introduced in [27]. For an introduction aimed for physicists, see [5, Sec. 5] or [29, Sec. 2].

[8]We have non-canonical isomorphisms $\text{Hom}(\text{Tors}\,A, U(1)) \simeq \text{Tors}\,A$ and $\text{Hom}(\text{Free}\,A, \mathbb{Z}) \simeq \text{Free}\,A$ and therefore $H^{d+1}(X;\mathbb{Z}) \simeq \text{Tors}\,H_d(X;\mathbb{Z}) \oplus \text{Free}\,H_{d+1}(X;\mathbb{Z})$ as Abelian groups.

Indeed, we can write

$$
\begin{array}{ccccccccc}
0 & \to & \mathrm{Ext}_{\mathbb{Z}}(H_d(X;\mathbb{Z}),\mathbb{Z}) & \xrightarrow{b} & H^{d+1}(X;\mathbb{Z}) & \xrightarrow{a} & \mathrm{Hom}_{\mathbb{Z}}(H_{d+1}(X;\mathbb{Z}),\mathbb{Z}) & \to & 0 \\
& & \Big\updownarrow & & \| & & \Big\downarrow & & \\
\cdots \xrightarrow{\pi_1} & & H^d(X;U(1)) & \xrightarrow{\beta} & H^{d+1}(X;\mathbb{Z}) & \xrightarrow{\iota} & H^{d+1}(X;\mathbb{R}) & \xrightarrow{\pi_2} & \cdots
\end{array}
\tag{2.20}
$$

such that

$$
\mathrm{Coker}\,\pi_1 = \mathrm{Ext}_{\mathbb{Z}}(H_d(X;\mathbb{Z}),\mathbb{Z}), \qquad \mathrm{Ker}\,\pi_2 = \mathrm{Hom}_{\mathbb{Z}}(H_{d+1}(X;\mathbb{Z}),\mathbb{Z}).
\tag{2.21}
$$

The homomorphism $\beta$ is the Bockstein; for $d = 1$, it maps a flat $U(1)$ connection to its first Chern class.

## 2.4 $4d$ WZW terms for $SU$ QCD ($N_f \geq 3$)

Let us now move on to the WZW term for $4d$ $SU$ QCD [3]. Somewhat surprisingly, a proper description of this term on general manifolds requires a slight extension of the ideas explained above, namely the use of the spin structure, as already emphasized in [18].

Consider $4d$ $SU(N_c)$ gauge theory with $N_f$ massless flavors of quarks. The low-energy limit is believed to be described by a sigma model whose target space is $SU(N_f)$. We first examine the case when $N_f \geq 3$. Given a field configuration $\sigma : M_4 \to SU(N_f)$, suppose we can pick an auxiliary five-dimensional manifold $W_5$ with $\partial W_5 = M_4$ and $\sigma : W_5 \to SU(N_f)$ suitably extended. Following our strategy explained above, we pick a closed five-form on the group manifold $SU(N_f)$, generating $H^5(SU(N_f);\mathbb{R}) \simeq \mathbb{R}$. There is a natural $SU(N_f)$-invariant one, which is $\mathrm{Tr}(\sigma^{-1}d\sigma)^5$. We then define the WZW term as $e^{ik\int_{W_5}\mathrm{Tr}(\sigma^{-1}d\sigma)^5}$, with a suitable coefficient $k$.

It was argued in [20] that the proper coefficient is given by

$$
e^{-S_{\mathrm{WZW}}[\sigma:M_4 \to SU(N_f)]} := \exp\left(2\pi i \cdot N_c \int_{W_5} \Gamma_5\right),
\tag{2.22}
$$

where

$$
\Gamma_{2n-1} := \left(\frac{i}{2\pi}\right)^n \frac{(n-1)!}{(2n-1)!} \mathrm{Tr}(\sigma^{-1}d\sigma)^{2n-1}
\tag{2.23}
$$

is normalized to integrate to 1 on the generator of $\pi_{2n-1}(SU(N_f)) \simeq \mathbb{Z}$ [30]. We will recall how the coefficient in (2.22) is fixed later in Sec. 4.2. Here we simply assume it is given.

Note that this is normalized against the generator of the homotopy group $\pi_5(SU(N_f)) \simeq \mathbb{Z}$, rather than that of the homology group $H_5(SU(N_f);\mathbb{Z}) \simeq \mathbb{Z}$. According to [31], the map

$$
\pi_{2n-1}(SU(N_f)) \to H_{2n-1}(SU(N_f);\mathbb{Z})
\tag{2.24}
$$

sends 1 to $(n-1)!$ times a generator. This means that, for $2n-1 = 3$, there is no difference between the normalization of the homotopy group and the homology group, and the expression $\exp(2\pi i k \int_{W_3} \Gamma_3)$ is exactly the $2d$ WZW term (2.11). On the other hand, for $2n-1 = 5$, $\Gamma_5$ above integrates to $1/2$ on the generator of $H_5(SU(N_f);\mathbb{Z})$, which in turn implies that, when $N_c$ is odd, the coupling (2.22) violates the quantization condition (2.14) discussed above based on (co)homology.

This can be confirmed explicitly for example by taking $\sigma : W_5 \to SU(N_f)$ to be the inclusion

$$
\sigma : \mathrm{Wu} \hookrightarrow SU(3) \subset SU(N_f),
\tag{2.25}
$$

where Wu is the Wu manifold[9] defined by

$$\text{Wu} = \{\sigma \in SU(3) \mid \sigma = \sigma^{\mathsf{T}}\} \simeq SU(3)/SO(3). \tag{2.26}$$

An explicit computation using differential forms[10] shows that $\int_{\text{Wu}} \Gamma_5 = 1/2$, and therefore the coupling (2.22) is ill-defined in the sense that it depends on the extension $\sigma : W_5 \to SU(N_f)$, not just on its boundary values.[11] The way out is to require spin structures in the manifolds which appear in the construction.

To understand how spin structures save the day, we need to recall some facts in algebraic topology, summarized in Appendix A. First, it is known that $H^*(SU(N_f); \mathbb{Z}) = \bigwedge_{\mathbb{Z}}[x_3, x_5, \cdots]$, where $x_i \in H^i(SU(N_f); \mathbb{Z})$. In particular, $x_5$ is the generator of $H^5(SU(N_f); \mathbb{Z}) \simeq \mathbb{Z}$. We abuse the notation and use the same symbol $x_i$ for the corresponding elements in $H^i(SU(N_f); \mathbb{R})$. We then have $\Gamma_5 = x_5/2$, since $\Gamma_5$ integrates to $1/2$ on the generator Wu of $H_5(SU(N_f); \mathbb{Z})$. We also denote the mod 2 reductions of $x_i$ by the same letter; it is known that $Sq^2 x_3 = x_5$ [32].

Then, for any closed manifold $W_5$ equipped with $\sigma : W_5 \to SU(N_f)$, one has

$$\int_{W_5} \sigma^*(x_5) = \int_{W_5} \sigma^*(Sq^2 x_3) = \int_{W_5} Sq^2 \sigma^*(x_3) = \int_{W_5} v_2(T) \sigma^*(x_3), \tag{2.27}$$

where all equalities are taken modulo 2 and we also used the fact $\int_M Sq^i a = \int_M v_i a$ for $\mathbb{Z}_2$-valued cohomology classes $a$, where $v_i(T)$ is the Wu class of the tangent bundle $T$. It is further known that $v_2(T) = w_2(T) + w_1(T)^2$. As we assume our $W_5$ to be oriented (i.e. $w_1(T) = 0$) and spin (i.e. $w_2(T) = 0$), the Wu class vanishes, $v_2(T) = 0$. This means $\int_{W_5} \sigma^*(x_5) \in 2\mathbb{Z}$ in our case, implying therefore $\int_{W_5} \sigma^*(\Gamma_5) \in \mathbb{Z}$. This makes the 4d WZW coupling (2.22) for odd $N_c$ well-defined on spin manifolds, at least when we can find a $W_5$ and $\sigma : W_5 \to SU(N_f)$ extending a given $\sigma : M_4 \to SU(N_f)$, where we assume both $M_4$ and $W_5$ are spin.

We now need to discuss whether such an extension really exists. This can be answered using the theory of bordisms. Given a manifold $X$, spin bordism groups $\Omega_d^{\text{spin}}(X)$ are defined as follows. We start from a closed spin manifold $M_d$ with a map $\sigma : M_d \to X$. We introduce an equivalence relation on such pairs $(M_d, \sigma)$ by saying that $\sigma : M_d \to X$ and $\sigma' : M_d' \to X$ are equivalent when there is a spin manifold $W_{d+1}$ and $\sigma : W_{d+1} \to X$ such that $\partial W_{d+1} = M_d \sqcup \overline{M_d'}$ and that $\sigma : W_{d+1} \to X$ extends both $\sigma : M_d \to X$ and $\sigma' : M_d' \to X$. The resulting equivalence classes form a group $\Omega_d^{\text{spin}}(X)$, where the group operation is given by a disjoint sum. If we require an orientation instead of a spin structure on $M_d$ and $W_{d+1}$, then we can similarly define the oriented bordism group $\Omega_d^{\text{oriented}}(X)$.

---

[9]It is a current standard practice to call this homogeneous space $SU(3)/SO(3)$ as the Wu manifold, as a quick google search https://www.google.com/search?q="wu+manifold" would abundantly show. It is however quite unclear whether the credit is correctly attributed. As we mention later, $SU(3)/SO(3)$ is a generator of $\Omega_5^{\text{oriented}}(pt) = \mathbb{Z}_2$. Another generator, $(S^1 \times \mathbb{CP}^2)/\mathbb{Z}_2$, where $\mathbb{Z}_2$ acts by a half-shift on $S^1$ and by the complex conjugation on $\mathbb{CP}^2$, was indeed discussed by Wu in [33], which was also the paper where he introduced the Wu classes. That $SU(3)/SO(3)$ is a generator was apparently first noticed by Calabi, as can be seen in [34]. On one hand, this latter paper still correctly attributed $SU(3)/SO(3)$ to Calabi and $(S^1 \times \mathbb{CP}^2)/\mathbb{Z}_2$ to Wu. On the other hand, a rather influential paper classifying simply-connected five-manifolds up to diffeomorphism [35] referred to a five-manifold $X_{-1}$ and called it the Wu manifold citing [36], which in turn referred to [33]. It does not seem to the authors, however, that the references [33,36] discussed $X_{-1}$. The paper [35] did not explicitly mention $SU(3)/SO(3)$, but a theorem of [35] says that if an isomorphism $H_2(M_5) \simeq H_2(M_5')$ preserves the linking pairing and $w_2$, then $M_5$ and $M_5'$ are diffeomorphic. From this theorem it is easy to conclude that $X_{-1} \simeq SU(3)/SO(3)$. This seems to be the origin of the name *Wu manifold* for $SU(3)/SO(3)$.

[10]For example, introduce a basis of $\mathfrak{su}(3)$ such that $\text{Tr}\,\lambda_a \lambda_b = 2\delta_{ab}$, so that $\lambda_{6,7,8}$ belongs to $\mathfrak{so}(3)$. Let $U = 1 + \lambda_a x^a \in SU(3)$ be an element close to the identity, and let $\sigma = UU^{\mathsf{T}}$. We find $\Gamma_5 = 4/(\sqrt{3}\pi^3)dx^1 \cdots dx^5$ at the origin. It is known that the volume of $SU(3)/SO(3)$ in this metric is $\sqrt{3}\pi^3/8$, see e.g. [37]. Therefore $\Gamma_5$ integrates to $1/2$.

[11]For example, given a $\sigma : W_5 \to SU(N_f)$ such that $\partial W_5 = M_4$, one can take $W_5' := W_5 \sqcup \text{Wu}$, with $\sigma$ on Wu given by the inclusion. This multiplies the value of (2.22) by $(-1)^{N_c}$.

We note that the bordism groups always split as

$$\Omega_d^{\text{spin}}(X) = \Omega_d^{\text{spin}}(pt) \oplus \widetilde{\Omega}_d^{\text{spin}}(X), \tag{2.28}$$

when $X$ is connected. Here, the first direct summand on the right hand side is the bordism group of a point and the second direct summand is known as the reduced bordism group. This split geometrically comes from the fact that any class $[\sigma : M_d \to X] \in \Omega_d^{\text{spin}}(X)$ determines $[M_d] \in \Omega_d^{\text{spin}}(pt)$ by forgetting the map $\sigma$, and $[M_d] \in \Omega_d^{\text{spin}}(pt)$ determines $[\sigma_0 : M_d \to X] \in \Omega_d^{\text{spin}}(X)$ where $\sigma_0$ sends $M_d$ to a single point on $X$.

We can show that the (reduced) bordism group $\widetilde{\Omega}_4^{\text{spin}}(SU(N_f))$ relevant for our question vanishes for $N_f \geq 3$; for details, see Appendix B.1.1. This means that any $\sigma : M_4 \to SU(N_f)$ is bordant to $\sigma_0 : M_4 \to SU(N_f)$ where $\sigma_0$ sends the entire spacetime to a single point. In other words, we can find a 5-manifold $W_5$ with $\partial W_5 = M_4 \sqcup \overline{M_4}$ such that there is a map $\sigma : W_5 \to SU(N_f)$ which extends $\sigma$ and $\sigma_0$ on both boundaries. We declare that the WZW term is trivial for the topologically trivial configuration $\sigma_0$. Then the WZW term for a nontrivial $\sigma$ is given by the integral (2.22) over $W_5$. This completes the specification of the 4$d$ WZW term for $N_f \geq 3$ for a general spin manifold $M_4$.

In passing, we note that it is known that Wu generates $\Omega_5^{\text{oriented}} = \mathbb{Z}_2$, which can be detected by the Stiefel-Whitney number $\int_W w_2(T) w_3(T)$, where $w_i(T)$ is the $i$-th Stiefel-Whitney class of the tangent bundle. In particular, $w_2$ is non-trivial on the Wu manifold, which means that it is not spin. That it is not spin can also be seen from our discussion above, where we showed that $x_5$ integrates to an even number on a spin manifold, while $x_5$ integrates to 1 on the Wu manifold.

## 2.5 4$d$ WZW terms for $SU$ QCD ($N_f = 2$)

Let us now consider the special case of $N_f = 2$. Since $\dim SU(2) = 3$, there is no appropriate five-form. Instead, we can show that $\widetilde{\Omega}_4^{\text{spin}}(SU(2)) = \mathbb{Z}_2$.[12] Using this, we can introduce the coupling

$$e^{-S_{\text{WZW}}[\sigma:M_4 \to SU(2)]} := (-1)^{N_c [\sigma:M_4 \to SU(2)]}, \tag{2.29}$$

where $[\sigma : M_4 \to SU(2)]$ is the equivalence class of this map $\sigma$ in $\widetilde{\Omega}_4^{\text{spin}}(SU(2)) = \mathbb{Z}_2$.

This sign has a more explicit description: a map $\sigma : M_4 \to SU(2)$ can be viewed as a collection of skyrmions. One way to define the worldline of cores of skyrmions is to define it as an inverse image of a point on $SU(2)$. After deforming $\sigma$ slightly if necessary, this inverse image is a collection of circles embedded in $M_4$. A point on $SU(2)$ is framed, and therefore the inverse image is also framed. As $M_4$ itself is assumed to be spin, this can be used to define spin structures on these circles. We can then assign a weight $\pm 1$ on each circle, depending on the spin structure, and we multiply them.[13] That this construction detects $\widetilde{\Omega}_4^{\text{spin}}(SU(2)) = \mathbb{Z}_2$ can be seen by studying the Atiyah-Hirzebruch spectral sequence (AHSS) computing it.

In the end, this means that a skyrmion behaves as a fermion if there is a non-trivial discrete WZW term. This fact was first explained more elementarily in [20]. The preceding discussions show that it is essential to have spin structures on $M_4$ to define the $SU(2)$ WZW term.

So far, we learned that the WZW terms for $SU$ QCD looked rather different depending on whether $N_f \geq 3$ (2.22) or $N_f = 2$ (2.29). Before proceeding, we would like to point out that there is in fact a close relationship between them. First, a map $\sigma : M_4 \to SU(2)$ can be thought

---

[12] This can be computed as in $N_f \geq 3$ using the AHSS, for which we refer the reader to Appendix B.1.1 for details. It also follows from the suspension isomorphism which says $\tilde{E}_d(S^n) = E_{d-n}(pt)$ for arbitrary generalized homology theories $E_\bullet$. As the spin bordism is a generalized homology theory and $SU(2) \simeq S^3$, the statement follows.

[13] Similar methods were used extensively in [38].

of as a map $\sigma : M_4 \to SU(N_f \geq 3)$ by composing with the standard inclusion $SU(2) \subset SU(N_f)$. Then we have an equality

$$(-1)^{N_c [\sigma : M_4 \to SU(2)]} = \exp\left( 2\pi i \cdot N_c \int_{W_5} \Gamma_5 \right), \tag{2.30}$$

where $W_5$ is a spin manifold such that $\partial W_5 = M_4$ and $\sigma$ is now extended as a map $\sigma : W_5 \to SU(N_f \geq 3)$. This equality was shown in [20] by explicitly constructing $W_5$ and $\sigma$, and then evaluating $\Gamma_5$ on it. This can also be shown using algebraic topology, see Appendix B.1.3.

Second, we can consider a spin$^c$ structure rather than a spin structure. Physically this corresponds to considering the non-chiral baryonic $U(1)$ charge of the fermions in the QCD. The relevant bordism group vanishes $\widetilde{\Omega}_4^{\text{spin}^c}(SU(2)) = 0$, see Appendix B.1.2. Therefore we can find a spin$^c$ manifold $W_5$ together with $\sigma : W_5 \to SU(2)$ such that $\sigma$ on $W_5$ extends $\sigma$ on $M_4 = \partial W_5$. We can then write the coupling

$$\exp\left( 2\pi i \cdot N_c \int_{W_5} \frac{F}{2\pi} \wedge \Gamma_3 \right). \tag{2.31}$$

Here, $F$ is the curvature of the $U(1)$ part of the spin$^c$ connection, or the background gauge field for the baryonic $U(1)$ symmetry, and $\Gamma_3$ was introduced in (2.23) and measures the skyrmion number. Once one allows $W_5$ to be spin$^c$, it is rather natural to introduce the spin$^c$ structure on the boundary $M_4$ itself. When $F = dA$, the expression above can be partially integrated to give

$$\exp\left( i \cdot N_c \int_{M_4} A \wedge \Gamma_3 \right), \tag{2.32}$$

which simply means that this term induces baryonic charge $N_c$ on a single skyrmion. In particular, because of the spin-charge relation imposed by the spin$^c$ structure, this term makes a skyrmion a fermion when $N_c$ is odd, implying at the same time that the expression (2.31) equals the $SU$ WZW term (2.29) for $N_f = 2$.

## 2.6  4$d$ WZW terms for $SO$ QCD ($N_f \geq 3$)

Let us next consider the quantization of the WZW term in $Spin(N_c)$ gauge theories. The $Spin(N_c)$ QCD contains fermions $\psi_{\alpha a i}$ where $\alpha = 1, 2$ is the spacetime spinor index, $a = 1, \ldots, N_c$ is the color index and $i = 1, \ldots, N_f$ is the flavor index. It is expected that, when $N_f$ is not too large with respect to $N_c$, the strongly-coupled dynamics generates the condensate

$$\Lambda^3 \sigma_{ij} := \langle \epsilon^{\alpha\beta} \delta^{ab} \psi_{\alpha a i} \psi_{\beta b j} \rangle, \tag{2.33}$$

where $\sigma_{ij}$ is a complex symmetric matrix and $\Lambda$ is the dynamical scale. $\sigma$ then takes values in the subset

$$\{ \sigma \in SU(N_f) \,|\, \sigma = \sigma^\mathsf{T} \} \subset SU(N_f), \tag{2.34}$$

which can be identified with the homogeneous space $SU(N_f)/SO(N_f)$.[14] In this section we assume $N_f \geq 3$.

Let us first discuss configurations $\sigma : M_4 \to SU(N_f)/SO(N_f)$ which can be extended to a map $\sigma : W_5 \to SU(N_f)/SO(N_f)$ such that $\partial W_5 = M_4$. We can now pull back the differential

---

[14]For more details on the identification of this subspace and the homogeneous space $SU(N_f)/SO(N_f)$, see [39, Appendix C].

form $\Gamma_5$ on $SU(N_f)$ to $SU(N_f)/SO(N_f)$, and define the WZW term as $e^{ik \int_{W_5} \Gamma_5}$ with a suitable coefficient $k$. As we will recall later in Sec. 4.2, the normalization coming from physics consideration is

$$\exp\left( 2\pi i \cdot N_c \int_{W_5} \frac{\Gamma_5}{2} \right). \tag{2.35}$$

We note that $\Gamma_5/2$ integrates to $1/4$ on the generator of $H_5(SU(N_f)/SO(N_f); \mathbb{Z}) \simeq \mathbb{Z}$, which can be taken to be the Wu manifold Wu. Therefore this coupling is not well-defined if we allow arbitrary oriented $W_5$.[15]

Stated differently, denoting the generator of $\mathbb{Z} \subset H^5(SU(N_f)/SO(N_f); \mathbb{Z})$ by $y_5$, we need to show that $\int_{W_5} y_5$ is a multiple of four once we impose some constraint on the allowed manifold $W_5$. As QCD requires spin structure, a natural condition to be imposed on $W_5$ is that it is endowed with spin structure. The argument (2.27) we used for $SU$ QCD can also be applied here, but it does not suffice, since it only shows that $\int_{W_5} y_5$ is even.

By a more detailed analysis, one can show that the integral on any spin manifold is in fact 0 mod 4 as required. We provide one method utilizing Adams spectral sequence in Appendix C,[16] and another method using $KO$-theory in Appendix D.1.

We now need to ask whether we can find such an extension $\sigma : W_5 \to SU(N_f)/SO(N_f)$. For brevity, we use the notation $X := SU(N_f)/SO(N_f)$ in the rest of this subsection. The relevant bordism group is computed in Appendix B.2.1, and is given by

$$\Omega_4^{\text{spin}}(X) = \underbrace{\Omega_4^{\text{spin}}(pt)}_{=\mathbb{Z}} \oplus \underbrace{\widetilde{\Omega}_4^{\text{spin}}(X)}_{=\begin{cases} \mathbb{Z}_2 \ (N_f \geq 5) \\ \mathbb{Z} \ (N_f = 4) \\ 0 \ (N_f = 3) \end{cases}} . \tag{2.36}$$

Let us first discuss the generic case of $N_f \geq 5$. We fix an explicit representative for the generator of $\Omega_4^{\text{spin}}(pt) = \mathbb{Z}$ and the generator of $\widetilde{\Omega}_4^{\text{spin}}(X) = \mathbb{Z}_2$. This fixes a representative for each element of $\Omega_4^{\text{spin}}(X)$. Any given $\sigma : M_4 \to X$ determines an element of $\Omega_4^{\text{spin}}(X)$, for which we already chose a specific representative $\underline{\sigma} : \underline{M_4} \to X$. Let us say that they are bordant via $\sigma : W_5 \to X$. Then we have the relation

$$e^{-S_{\text{WZW}}[\sigma:M_4 \to X]} = \exp\left( 2\pi i \cdot N_c \int_{W_5} \frac{\Gamma_5}{2} \right) e^{-S_{\text{WZW}}[\underline{\sigma}:\underline{M_4} \to X]}. \tag{2.37}$$

Therefore we are left to define the values of $e^{-S_{\text{WZW}}[\underline{\sigma}:\underline{M_4} \to X]}$ for the chosen representatives of generators of $\Omega_4^{\text{spin}}(X)$.

For the generator of $\Omega_4^{\text{spin}}(pt) = \mathbb{Z}$, we simply declare that $e^{-S_{\text{WZW}}} = 1$. For the generator of $\widetilde{\Omega}_4^{\text{spin}}(X) = \mathbb{Z}_2$, there is a subtler issue we need to deal with. For a chosen representative $\underline{\sigma} : \underline{M_4} \to X$ for the generator, we note that twice the generator is null-bordant such that $\partial W_5$ consists of two copies of $\underline{\sigma} : \underline{M_4} \to X$. This implies that[17]

$$\left( e^{-S_{\text{WZW}}[\underline{\sigma}:\underline{M_4} \to X]} \right)^2 = \exp\left( 2\pi i \cdot N_c \int_{W_5} \frac{\Gamma_5}{2} \right), \tag{2.38}$$

---

[15]When $N_c$ is even, $(-1)^F$ in the spacetime $Spin$ group and $-1$ in $SO(N_c)$ act in the same manner on the fermions. This allows us to put the $SO$ QCD on non-spin oriented manifolds, as we will detail in Sec. 3.3. In this case the consistency of the WZW term is realized in a subtler way. We come back to this question in the last part of this paper in Sec. 5.2.2.

[16]See in particular the footnote 33.

[17]In [40, Sec. 4], a nice representative $\underline{\sigma} : \underline{M_4} \to X$ was constructed, where $\underline{M_4} = T^2 \times T^2$ and the configuration $\underline{\sigma}$ is invariant under the spacetime parity transformation. This guarantees that the right hand side of (2.38) is trivial, and $e^{-S_{\text{WZW}}[\underline{\sigma}:\underline{M_4} \to X]} = \pm 1$.

for which there are two solutions differing by a sign.

Let us pick a particular solution to (2.38). Our consideration so far is sufficient to define *a* WZW term

$$e^{-S_{\text{WZW}}^{(1)}[\sigma:M_4 \to X]}, \tag{2.39}$$

for all configurations. Another solution to (2.38) then leads to

$$e^{-S_{\text{WZW}}^{(2)}[\sigma:M_4 \to X]} = e^{-S_{\text{WZW}}^{(1)}[\sigma:M_4 \to X]} \cdot \chi\Big([\sigma : M_4 \to X]\Big), \tag{2.40}$$

where $\chi$ is a non-trivial character

$$\chi \in \text{Hom}(\widetilde{\Omega}_4^{\text{spin}}(X), U(1)) = \mathbb{Z}_2. \tag{2.41}$$

We emphasize that neither $S_{\text{WZW}}^{(1)}$ nor $S_{\text{WZW}}^{(2)}$ is privileged at this point. The solutions to (2.38) form an affine space, or a torsor, over $\text{Hom}(\widetilde{\Omega}_4^{\text{spin}}(X), U(1)) = \mathbb{Z}_2$.

The path integral of $SO$ QCD should provide one specific solution among them. Determining it in any meaningful manner would be an interesting question. We provide a tentative way forward in Appendix D.2, leaving its precise implementation to the future.

The non-trivial character (2.41) can be explicitly determined and is given by

$$\chi\Big([\sigma : M_4 \to X]\Big) = \exp\left( 2\pi i \int_{M_4} \frac{1}{2} \mathcal{P}(w_2(\sigma)) \right), \tag{2.42}$$

where $w_2$ is the generator of $H^2(SU(N_f)/SO(N_f); \mathbb{Z}_2)$, $w_2(\sigma) := \sigma^*(w_2)$ is its pull-back by the scalar field, and $\mathcal{P}$ is the Pontrjagin square. For the derivation, see Appendix B.4. As $w_2(\sigma)$ represents the worldsheet of the electric flux tube, the character above adds a factor of $-1$ at each intersection of two flux tubes.

Let us now discuss the non-generic cases $N_f = 3, 4$. For $N_f = 3$, $\widetilde{\Omega}_4^{\text{spin}}(X) = 0$ and therefore the WZW term is completely specified at this point as in Sec. 2.4. For $N_f = 4$, we can assign an arbitrary phase $e^{-S_{\text{WZW}}} = e^{i\theta}$ for the generator of $\widetilde{\Omega}_4^{\text{spin}}(X) = \mathbb{Z}$. As $\theta$ can be continuously deformed, it does not affect the deformation class of the WZW term, but the actual WZW term depends on the value of $\theta$, and should be fixed by the QCD path integral. Similarly as in $N_f \geq 5$, the choice $\theta = 0$ or $\pi$ is not privileged at this point, since $\theta$ depends on the choice of the particular representative of the generator. Fixing this ambiguity is an interesting question left to the future.[18]

## 2.7  4d WZW terms for $SO$ QCD ($N_f = 2$)

The $SO$ WZW term for $N_f = 2$ is also interesting. This time, the sigma model target space is $SU(2)/SO(2) \simeq S^2$, for which we find that $\widetilde{\Omega}_4^{\text{spin}}(S^2) = \mathbb{Z}_2$[19]. This allows us to write down a WZW term

$$(-1)^{N_c [\sigma:M_4 \to S^2]}, \tag{2.43}$$

where $[\sigma : M_4 \to S^2]$ is the reduced bordism class in $\widetilde{\Omega}_4^{\text{spin}}(S^2) = \mathbb{Z}_2$.

The bordism invariant with odd $N_c$ has the following mathematical interpretation. We perturb $\sigma$ slightly to make it generic. Then we take the inverse image of a point on $S^2$ under $\sigma$. This defines a union of surfaces $\Sigma$ within $M_4$. Since a point in $S^2$ is obviously framed, its inverse image is also framed. Given a spin structure on $M_4$, this framing induces a spin structure on $\Sigma$. We then take the Arf invariant of the spin surface $\Sigma$. Physically, the inverse image of a point on $S^2$ is the color flux tube associated to $\pi_2(S^2) = \mathbb{Z}$, and the 2d effective

---

[18]A proposal was recently given in [40, Sec. 4], which uses the gauged WZW term in an essential manner.

[19]It follows from the suspension isomorphism, as discussed in footnote 12.

theory on this flux tube is the non-trivial fermionic invertible phase corresponding to the non-trivial element of $\text{Hom}(\Omega_2^{\text{spin}}(pt), U(1)) = \mathbb{Z}_2$, which is known as the Arf theory or the Kitaev chain.

The boundary of the Arf theory famously carries an odd number of Majorana fermion zero modes. Therefore, this means that the boundary of the electric flux tube with odd/even $N_c$ carries an odd/even number of Majorana zero modes. To see this in the ultraviolet description, recall that the electric flux tube of the $Spin(N_c)$ gauge theory carries an electric flux in the spinor representation, and therefore ends on the Wilson line in the spinor representation. In our $Spin(N_c)$ QCD, the dynamical fermions $\psi_a$ are in the vector representation of $Spin(N_c)$, with the index $a$ running from 1 to $N_c$. The Wilson line in the spinor representation of $Spin(N_c)$ then has an action of the gamma matrices $\Gamma_a$ ($a = 1, \ldots, N_c$), which can also be considered as Majorana fermions, and there are clearly $N_c$ of them.

We also note that the map $\widetilde{\Omega}_4^{\text{spin}}(S^2) = \mathbb{Z}_2 \to \widetilde{\Omega}_4^{\text{spin}}(SU(N_f \geq 3)/SO(N_f))$ is a zero map, as shown in Appendix B.2.1. This means that the $SO$ WZW term for $N_f = 2$ can be expressed as an $SO$ WZW term for $N_f \geq 3$ using (2.35). We can show that

$$(-1)^{N_c[\sigma:M_4 \to S^2]} = \exp\left( 2\pi i \cdot N_c \int_{W_5} \frac{\Gamma_5}{2} \right), \tag{2.44}$$

where $W_5$ is found by considering $N_f = 2$ configurations as $N_f \geq 3$ configurations. This equality can be shown using algebraic topology, see Appendix B.2.2. The two-fold ambiguity of the $N_f \geq 3$ WZW term given by (2.42) is immaterial here, since $\mathcal{P}(w_2) = 0$ on $S^2$ simply because $H^4(S^2)$ is trivial.

## 2.8 Invertible phases and the Anderson dual of the bordism group

We can draw some lessons from the preceding analyses of 4$d$ WZW terms, and revise our general discussion in Sec. 2.3. Suppose we have a $d$-dimensional spin theory with a scalar field taking values in a manifold $X$. Here by a spin theory we mean that the spacetime manifold $M_d$ is equipped with a spin structure. We would like to specify a $U(1)$-valued phase $e^{-S[\phi:M_d \to X]}$ in the exponentiated action. Physics imposes various consistency conditions on such a $U(1)$-valued phase. Consistent such phases are now commonly called *invertible phases*; when we require spin structures on spacetime manifolds, they are *spin invertible phases*.

### 2.8.1 Free part

As before, we assume that there is a closed $(d+1)$-form field strength $G$, such that when the scalar field $\phi : M_d \to X$ is extensible to $\phi : W_{d+1} \to X$ with $\partial W_{d+1} = M_d$, the coupling is given by

$$e^{-S[\phi:M_d \to X]} = e^{i \int_{\phi(W_{d+1})} G}. \tag{2.45}$$

Here we allow $G$ to consist not only of differential forms on $X$ but also of the Pontrjagin classes $p_i$ of $\phi(W_{d+1})$. Since $\mathbb{Q}[p_1, p_2, \ldots] = H^*(BSpin; \mathbb{Q})$, this means that we regard $G$ as an element of $H^{d+1}(BSpin \times X; \mathbb{Q})$.

For this coupling to be independent of the extension, we require that

$$\int_{\phi(W_{d+1})} G \in 2\pi\mathbb{Z}, \tag{2.46}$$

for all maps from a closed spin manifold $\phi : W_{d+1} \to X$. The pairing (2.46) determines a homomorphism

$$\Omega_{d+1}^{\text{spin}}(X) \to \mathbb{Z}, \tag{2.47}$$

as opposed to our discussion in (2.14) where we had the ordinary homology group $H_{d+1}(X;\mathbb{Z})$ instead of the bordism group $\Omega_{d+1}^{\text{spin}}(X)$. Such a homomorphism is specified by an element of

$$\text{Hom}_{\mathbb{Z}}(\Omega_{d+1}^{\text{spin}}(X),\mathbb{Z}). \tag{2.48}$$

Here, we note that

$$\text{Hom}_{\mathbb{Z}}(\Omega_{d+1}^{\text{spin}}(X),\mathbb{Z}) \to \text{Hom}_{\mathbb{Z}}(\Omega_{d+1}^{\text{spin}}(X),\mathbb{Z}) \otimes \mathbb{Q} = H^{d+1}(BSpin \times X;\mathbb{Q}) \tag{2.49}$$

is an injection, because $\Omega_{\bullet}^{\text{spin}}(pt) \otimes \mathbb{Q} = H_{\bullet}(BSpin;\mathbb{Q})$. The discussion thus far generalizes the $4d$ WZW terms for $SU$ QCD with $N_f \geq 3$.

### 2.8.2 Torsion part

We can also consider the case when $G$ vanishes. In this case, the relation (2.45) means that the value $e^{-S[\phi:M_d \to X]}$ only depends on the bordism class of $\phi : M_d \to X$, which we denote by $[\phi : M_d \to X] \in \Omega_d^{\text{spin}}(X)$. We then see that the coupling is given by

$$e^{-S[\phi:M_d \to X]} = \chi\Big([\phi : M_d \to X]\Big), \tag{2.50}$$

where $\chi$ is now a character

$$\chi : \Omega_d^{\text{spin}}(X) \to U(1). \tag{2.51}$$

Again this is different from what we had in (2.16) where we encountered $H_d(X;\mathbb{Z})$ instead of $\Omega_d^{\text{spin}}(X)$. Such characters up to continuous deformation are classified by

$$\text{Hom}(\text{Tors}\,\Omega_d^{\text{spin}}(X),U(1)) = \text{Ext}_{\mathbb{Z}}(\Omega_d^{\text{spin}}(X),\mathbb{Z}). \tag{2.52}$$

The discussion generalizes the $4d$ WZW terms for $SU$ QCD with $N_f = 2$.

### 2.8.3 Combining the two and the Anderson dual

A general invertible phase is a certain combination of these two extremes, as we saw in the case of $SO$ WZW terms. There, the consideration of cases where $\phi : M_4 \to X$ extends to $\phi : W_5 \to X$ only determined the invertible phase up to a multiplication by a character (2.51).

This means that the group $\text{Inv}_{\text{spin}}^d(X)$ of deformation classes of $d$-dimensional spin invertible phases sits in a short exact sequence

$$0 \to \text{Ext}_{\mathbb{Z}}(\Omega_d^{\text{spin}}(X),\mathbb{Z}) \longrightarrow \text{Inv}_{\text{spin}}^d(X) \overset{a}{\longrightarrow} \text{Hom}_{\mathbb{Z}}(\Omega_{d+1}^{\text{spin}}(X),\mathbb{Z}) \to 0. \tag{2.53}$$

Note the difference with respect to (2.17), where we had homology groups instead of bordism groups. As before, for a class $c \in \text{Inv}_{\text{spin}}^d(X)$, $a(c)$ specifies the pairing (2.46). When $a(c)$ vanishes, the invertible phase is continuously deformable to a flat one, which is then given by a character (2.51).

The explanations thus far should have clarified why one needs to use (co)bordisms instead of (co)homologies. In the end, in many contexts the spin QFT only deals with spacetimes equipped with spin structure, and does not deal with arbitrary representatives of homology classes which can be unoriented, non-spin, or even not expressible as an image from manifolds. Without doubt the homology groups are the easiest algebraic-topological invariants of spaces, but they are not perfectly natural for the purpose of spin QFT.

Let us discuss more about the sequence (2.53). Mathematically, $\Omega_\bullet^{\mathrm{spin}}(X)$ is an example of generalized homology theory. For any generalized homology theory $E_\bullet(X)$, there is the so-called *Anderson dual*[20] cohomology theory $(DE)^\bullet(X)$, satisfying

$$0 \to \mathrm{Ext}_{\mathbb{Z}}(E_d(X), \mathbb{Z}) \longrightarrow (DE)^{d+1}(X) \longrightarrow \mathrm{Hom}_{\mathbb{Z}}(E_{d+1}(X), \mathbb{Z}) \to 0. \tag{2.54}$$

The universal coefficient theorem (2.17) for ordinary homology $H(-;\mathbb{Z})$ means that $DH(-;\mathbb{Z}) = H(-;\mathbb{Z})$. It is also known that the complex K-theory satisfies $DK = K$ and the real K-theory satisfies $DKO = KSp$.

Comparing (2.54) with (2.53), we conclude that

$$\mathrm{Inv}_{\mathrm{spin}}^d(X) = (D\Omega_{\mathrm{spin}})^{d+1}(X). \tag{2.55}$$

This is the meaning of the recently often-found remark that invertible phases are classified by the Anderson dual of the bordism group whose degree is shifted by one, originally formulated in [12].

The Anderson dual of the bordism group describes the deformation classes of the invertible phases. The invertible phases themselves, not their deformation classes, should be described by the differential version of the Anderson dual of the bordism group. We do not get into its details in this paper.

### 2.8.4 $SU$ WZW terms

Let us now re-examine the case when $d = 4$ and $X = SU(N_f)$. First consider the case $N_f \geq 3$. In this case we have

$$0 \to \underbrace{\mathrm{Ext}_{\mathbb{Z}}(\Omega_4^{\mathrm{spin}}(SU(N_f)), \mathbb{Z})}_{=0} \to (D\Omega_{\mathrm{spin}})^5(SU(N_f)) \to \underbrace{\mathrm{Hom}_{\mathbb{Z}}(\Omega_5^{\mathrm{spin}}(SU(N_f)), \mathbb{Z})}_{=\mathbb{Z}} \to 0, \tag{2.56}$$

showing that

$$(D\Omega_{\mathrm{spin}})^5(SU(N_f)) \simeq \mathrm{Hom}_{\mathbb{Z}}(\Omega_5^{\mathrm{spin}}(SU(N_f)), \mathbb{Z}) \simeq \mathbb{Z}, \tag{2.57}$$

whose elements are labeled by $N_c$.

Note that

$$\Omega_5^{\mathrm{spin}}(SU(N_f)) \simeq \mathbb{Z} \to H_5(SU(N_f); \mathbb{Z}) \simeq \mathbb{Z} \tag{2.58}$$

is a multiplication by two. Therefore, dually,

$$H^5(SU(N_f); \mathbb{Z}) \simeq \mathbb{Z} \to \mathrm{Hom}_{\mathbb{Z}}(\Omega_5^{\mathrm{spin}}(SU(N_f)), \mathbb{Z}) \simeq \mathbb{Z} \tag{2.59}$$

is also a multiplication by two. This corresponded to the fact that the generator $x_5$ of $H^5(SU(N_f); \mathbb{Z})$ integrates to even integers on the image of spin manifolds in $SU(N_f)$, which can be seen from $x_5 = Sq^2 x_3$, as in (2.27). This allows us to put (2.59) into a short exact sequence

$$0 \to \underbrace{H^5(SU(N_f); \mathbb{Z})}_{\mathbb{Z}} \to \underbrace{(D\Omega_{\mathrm{spin}})^5(SU(N_f))}_{\mathbb{Z}} \to \underbrace{H^3(SU(N_f); \mathbb{Z}_2)}_{\mathbb{Z}_2} \to 0. \tag{2.60}$$

As we will analyze in detail in Appendix B.1.3, this sequence naturally arises from the analysis of the Atiyah-Hirzebruch spectral sequence determining $(D\Omega_{\mathrm{spin}})^5(SU(N_f))$.

When $N_f = 2$, the sequence (2.56) is modified to

$$0 \to \underbrace{\mathrm{Ext}_{\mathbb{Z}}(\Omega_4^{\mathrm{spin}}(SU(2)), \mathbb{Z})}_{=\mathbb{Z}_2} \to (D\Omega_{\mathrm{spin}})^5(SU(2)) \to \underbrace{\mathrm{Hom}_{\mathbb{Z}}(\Omega_5^{\mathrm{spin}}(SU(2)), \mathbb{Z})}_{=0} \to 0, \tag{2.61}$$

---

[20]The concept of the Anderson dual goes back to [41]. $DE$ is also often denoted as $I_{\mathbb{Z}}E$.

and the sequence (2.60) is modified to

$$0 \to \underbrace{H^5(SU(2); \mathbb{Z})}_{0} \to \underbrace{(D\Omega_{\mathrm{spin}})^5(SU(2))}_{\mathbb{Z}_2} \to \underbrace{H^3(SU(2); \mathbb{Z}_2)}_{\mathbb{Z}_2} \to 0. \qquad (2.62)$$

Comparing (2.60) and (2.62), we find that the pull-back along $SU(2) \subset SU(N_f \geq 3)$ is the mod 2 reduction

$$(D\Omega_{\mathrm{spin}})^5(SU(N_f)) \simeq \mathbb{Z} \to (D\Omega_{\mathrm{spin}})^5(SU(2)) \simeq \mathbb{Z}_2. \qquad (2.63)$$

This explains the relation (2.30) from a different, more abstract point of view.

### 2.8.5 $SO$ WZW terms

Let us next examine the $SO$ WZW terms, for which $X = SU(N_f)/SO(N_f)$. First consider $N_f \geq 3$. We have

$$0 \to \underbrace{\mathrm{Ext}_{\mathbb{Z}}(\Omega_4^{\mathrm{spin}}(X), \mathbb{Z})}_{=\mathbb{Z}_2 \text{ or } 0} \to (D\Omega_{\mathrm{spin}})^5(X) \to \underbrace{\mathrm{Hom}_{\mathbb{Z}}(\Omega_5^{\mathrm{spin}}(X), \mathbb{Z})}_{=\mathbb{Z}} \to 0, \qquad (2.64)$$

and the WZW term for the $SO(N_c)$ QCD corresponds to $N_c$ times the generator of the free part $\mathbb{Z}$. We also note that

$$H^5(SU(N_f)/SO(N_f); \mathbb{Z}) \supset \mathbb{Z} \to \mathrm{Hom}_{\mathbb{Z}}(\Omega_5^{\mathrm{spin}}(SU(N_f)/SO(N_f)), \mathbb{Z}) \simeq \mathbb{Z}, \qquad (2.65)$$

is a multiplication by four. This cannot be understood by just noting that the generator $y_5$ of $H^5(SU(N_f)/SO(N_f); \mathbb{Z})$ is in the image of $Sq^2$ as before. This time, the multiplication by four comes from the extension by $H^3(SU(N_f)/SO(N_f); \mathbb{Z}_2) = \mathbb{Z}_2$ and then another extension by $H^2(SU(N_f)/SO(N_f); \mathbb{Z}_2) = \mathbb{Z}_2$, as can be seen from the analysis of the Atiyah-Hirzebruch spectral sequence, see Appendix B.2.2. It can also be understood using the Adams spectral sequence, see Appendix C.

We have not succeeded in determining the torsion part of the WZW term of the $SO$ QCD. We discuss a tentative way forward in Appendix. D.2.

When $N_f = 2$ we instead have $(D\Omega_{\mathrm{spin}})^5(SU(2)/SO(2)) = \mathbb{Z}_2$. The pull-back

$$(D\Omega_{\mathrm{spin}})^5(SU(N_f)/SO(N_f)) \simeq \mathbb{Z} \to (D\Omega_{\mathrm{spin}})^5(SU(2)/SO(2)) \simeq \mathbb{Z}_2 \qquad (2.66)$$

is the mod 2 reduction.

### 2.8.6 Freed-Gu-Wen phases

Before proceeding, we note that in an earlier paper [18] by Freed, the WZW terms are analyzed using a generalized cohomology theory which was called $E^\bullet$ in the paper, not by $(D\Omega_{\mathrm{spin}})^\bullet$. This theory $E^\bullet$ is obtained by keeping the first two non-trivial group $E^0(pt) = (D\Omega_{\mathrm{spin}})^0(pt) = \mathbb{Z}$ and $E^2(pt) = (D\Omega_{\mathrm{spin}})^2(pt) = \mathbb{Z}_2$ while killing all non-trivial $(D\Omega_{\mathrm{spin}})^{\bullet > 2}(pt)$ so that $E^{\bullet > 2}(pt) = 0$. This essentially means that $E^\bullet$ only captures spin invertible phases understandable by a single action of $Sq^2$.

The same class of invertible phases was also studied in the condensed matter literature by Gu and Wen in [42], where the cohomology theory $E^\bullet$ was called *supercohomology*. These circumstances made this class of invertible phases to be known under various names in the literature, namely Freed-Gu-Wen phases, Gu-Wen phases, or supercohomology phases. The analysis above shows that the WZW terms for $SU(N_c)$ QCD are an example of Freed-Gu-Wen phases, whereas the WZW terms for $SO(N_c)$ QCD goes beyond this class.

# 3 Interlude: the anomaly in the ultraviolet

Before going to the discussion of the gauged WZW terms, we need to discuss the anomalies of QCD in the ultraviolet.

## 3.1 Generalities

Let us first recall the modern characterization of anomalies. An anomalous QFT with flavor symmetry $G$ in $d$ spacetime dimensions is considered as realized on the boundary of an invertible theory with $G$ symmetry in $d + 1$ spacetime dimensions; the deformation class $\alpha \in \mathrm{Inv}_{\mathrm{spin}}^{d+1}(BG) = (D\Omega_{\mathrm{spin}})^{d+2}(BG)$ of the invertible theory is the anomaly of the theory on the boundary. Here and in the following we assume that the spacetime is equipped with spin structure.

Using the short exact sequence (2.53), we can extract the quantity

$$a(\alpha) \in \mathrm{Hom}_{\mathbb{Z}}(\Omega_{d+2}^{\mathrm{spin}}(BG), \mathbb{Z}), \tag{3.1}$$

which is the anomaly polynomial. The anomaly polynomial is usually given as an element of

$$\mathrm{Hom}_{\mathbb{Z}}(\Omega_{d+2}^{\mathrm{spin}}(BG), \mathbb{Z}) \otimes \mathbb{Q} \simeq H^{d+2}(BSpin \times BG; \mathbb{Q}), \tag{3.2}$$

which, as we saw, is given by a polynomial of spacetime Pontrjagin classes and the differential forms on $BG$. Furthermore, for a chiral fermion in the representation $V$ of $G$, the anomaly polynomial is given by

$$a(\alpha(V)) = (\hat{A}\,\mathrm{ch}(V))_{d+2}, \tag{3.3}$$

where $\hat{A} \in H^*(BSpin)$ is the A-roof polynomial and $\mathrm{ch}(V)$ is the Chern character of the representation $V$; we remind the reader that $\mathrm{ch}(V)$ can be written via the Chern-Weil homomorphism as $\mathrm{Tr}_V e^{iF/(2\pi)}$ where $F$ is the curvature of the $G$-bundle. More generally, the anomaly of a fermion system is given by the $\eta$ invariant, see [43] and references therein. This carries not only the data of the anomaly polynomial but also the torsion part.

Given a fermion system charged under $G'$, we would like to gauge its normal subgroup $G \subset G'$. This requires $G$ to be anomaly free. The flavor symmetry group is then $F = G'/G$, if there is no mixed anomaly between $G$ and $F$, that is, *if* the anomaly $\alpha_{G'} \in \mathrm{Inv}_{\mathrm{spin}}^{d+1}(BG')$ of the fermion system is pulled back from an element $\alpha_F \in \mathrm{Inv}_{\mathrm{spin}}^{d+1}(BF)$ via the projection $p : G' \to F$, $\alpha_{G'} = p^*(\alpha_F)$. Then, the anomaly of the gauged theory under $F$ is simply given by $\alpha_F$.

In the presence of the mixed anomaly, it is not even guaranteed that $F$ is the flavor symmetry group [44]. In this paper we only consider the simpler cases where there is no mixed anomaly in the original ungauged fermion system.

## 3.2 *SU* QCD

Let us first consider the 4d $SU(N_c)$ QCD with $N_f \geq 3$ flavors. The fermions are charged under $G' = SU(N_c) \times SU(N_f) \times SU(N_f)'$ where the prime is to distinguish two flavor symmetry factors.

The chiral fermions are in the representations $V \otimes \bar{W}$ and $W \otimes \bar{V}'$, where $V$, $V'$ and $W$ are the fundamental representations of $SU(N_f)$, $SU(N_f)'$ and $SU(N_c)$ respectively. The anomaly polynomial of the fermions in $V \otimes \bar{W}$ is simply $N_c\,\mathrm{ch}\,V - N_f\,\mathrm{ch}\,W$, and that of the fermions in $W \otimes \bar{V}'$ is $N_f\,\mathrm{ch}\,W - N_c\,\mathrm{ch}\,V'$. As $\Omega_5^{\mathrm{spin}}(BSU(N_f) \times BSU(N_c)) = 0$, the anomaly polynomial completely determines the anomaly. Combining, the total anomaly of the fermion system before gauging is

$$N_c(\mathrm{ch}\,V - \mathrm{ch}\,V') = \frac{N_c}{2}(c_3 - c_3'), \tag{3.4}$$

where $c_i \in H^{2i}(BSU(N_f); \mathbb{Z})$ and $c_i' \in H^{2i}(BSU(N_f)'; \mathbb{Z})$ denote the Chern classes.

We are going to gauge $G = SU(N_c)$. The quotient $G'/G$ is $F = SU(N_f) \times SU(N_f)'$, and the anomaly (3.4) above is clearly pulled back from the anomaly of the quotient group $F$. Therefore, there is no mixed anomaly, the flavor symmetry of the gauge theory is $F$, and its anomaly is given by (3.4).

Note that the anomaly polynomial has a fractional coefficient in front of $c_3 - c_3'$, which therefore is not an element of $H^6(BSU(N_f); \mathbb{Z})$. The expression (3.4) still integrates to an integer on a spin manifold, as it arises via the index theorem applied to this fermion bundle. Another explanation can be given using the fact that $c_3 = Sq^2 c_2$ modulo 2, which guarantees that the integral of $c_3$ on a spin manifold to be even.

We can also consider the $U(1)$ baryon number symmetry, which assigns charge $+1$, $-1$ to the chiral fermions charged under $SU(N_f)$, $SU(N_f)'$ respectively. The contribution to the anomaly polynomial can be similarly obtained and is given by

$$N_c \left[\frac{F}{2\pi}\right](c_2 - c_2'),  \tag{3.5}$$

where $[F/(2\pi)]$ is the first Chern class of the $U(1)$ bundle. As $-1 \in U(1)$ acts on the fermions in the same way as the 360° rotation on the fermions, the spin structure of the spacetime can be upgraded to the spin$^c$ structure.

### 3.3  *SO* **QCD**

Let us next consider the 4d $SO(N_c)$ QCD with $N_f \geq 3$ flavors. The chiral fermions are in the representation $V \otimes W$ where $V$, $W$ are the fundamental representations of $SU(N_f)$ and $SO(N_c)$ respectively. The anomaly polynomial is simply given by

$$\frac{N_c}{2} c_3  \tag{3.6}$$

as before. Since $\Omega_5^{\mathrm{spin}}(BSO \times BSU) = 0$ as we compute in Appendix B.4, this is sufficient to completely specify the anomaly of the fermion system. This anomaly is pulled back from the anomaly of $BSU$. Therefore, there is no mixed anomaly, the flavor symmetry is $SU(N_f)$, and the anomaly of the QCD is given by (3.6).

We also note that the $SO(N_c)$ gauge theory alone possesses the magnetic $\mathbb{Z}_2$ 1-form symmetry [24, 45], and can be coupled to its background gauge field $B \in H^2(M_4; \mathbb{Z}_2)$ via

$$\exp\left(2\pi i \cdot \frac{1}{2} \int_{M_4} B w_2(c)\right),  \tag{3.7}$$

where $c$ is the $SO(N_c)$ gauge bundle over the spacetime $M_4$.

In the following, we concentrate on the case when $N_c = 2n_c$ is even. The following discussions also depend on whether we have the $SO(N_c)_+$ theory or the $SO(N_c)_-$ theory [45], distinguished by whether the coupling $2\pi i \int_{M_4} \mathcal{P}(w_2(c))/2$ is absent or not, where $\mathcal{P}$ is the Pontrjagin square. Here we only consider the $SO(N_c)_+$ theory.

We would now like to take into account that the center $\mathbb{Z}_2$ actions of gauge $SO(2n_c)$ and the spacetime $Spin$ group on matter fermions are identical, and therefore can be identified. The fermion is now charged under

$$[Spin(\text{spacetime}) \times SO(2n_c)]/\mathbb{Z}_2 \times SU(N_f).  \tag{3.8}$$

It has a subgroup $SO(2n_c) = [\{1, (-1)^F\} \times SO(2n_c)]/\mathbb{Z}_2$ which we are going to gauge. As computed in Appendix B.4, all possible anomalies under the structure (3.8) are pull-backs from

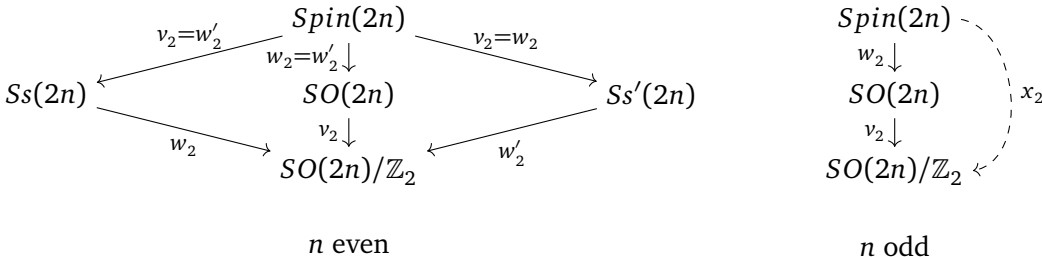

Figure 1: Various quotients of $Spin(2n)$ and the characteristic classes associated to their bundles. Solid arrows are $\mathbb{Z}_2$ quotients and the dashed arrow is a $\mathbb{Z}_4$ quotient. Furthermore, each arrow is labeled by the corresponding obstruction classes.

either $[Spin(\text{spacetime}) \times SO(2n_c)]/\mathbb{Z}_2$ or $SU(N_f)$. The fermion can be made massive under the former, so the anomaly restricted there is zero. Therefore, the anomaly of the fermion system under (3.8) is a pull-back from the $SU(N_f)$ anomaly specified by (3.6). After the gauging, therefore, we have the spacetime structure

$$Spin(\text{spacetime})/\mathbb{Z}_2 \times SU(N_f) = SO(\text{spacetime}) \times SU(N_f), \tag{3.9}$$

and the anomaly of the gauged theory is specified by (3.6).

There is, in addition, a mixed anomaly between the one-form symmetry and the spacetime symmetry, as pointed out recently in [25]. We will now review this mixed anomaly. Our discussion requires various obstruction classes associated to quotients of spin groups, which are summarized in Fig. 1.

We first note that since the fermions are charged under (3.8), whether the $SO(2n_c)/\mathbb{Z}_2$ gauge bundle lifts to an $SO(2n_c)$ bundle is synchronized with whether the spacetime tangent bundle lifts to a spin bundle. In terms of cohomology classes, this means

$$v_2(c) = w_2(T), \tag{3.10}$$

where $v_2(c) \in H^2(M_4; \mathbb{Z}_2)$ measures the obstruction to the lifting of the $SO(2n_c)/\mathbb{Z}_2$ gauge bundle, and $w_2(T) \in H^2(M_4; \mathbb{Z}_2)$ is the Stiefel-Whitney class of the tangent bundle specifying whether it lifts to a spin bundle.

For odd $n_c$, $\pi_1(SO(2n_c)/\mathbb{Z}_2) \simeq \mathbb{Z}_4$, and there is correspondingly a $\mathbb{Z}_4$-valued cocycle $x_2(c)$ for a given gauge bundle. Its mod 2 reduction is the class $v_2(c)$ controlling the lift from $SO(2n_c)/\mathbb{Z}_2$ to $SO(2n_c)$. Furthermore, if the $SO(2n_c)/\mathbb{Z}_2$ bundle actually comes from an $SO(2n_c)$ bundle, $x_2(c)$ is given by sending the mod 2 cocycle $w_2(c)$ by $\mathbb{Z}_2 \to \mathbb{Z}_4$ which sends 1 (mod 2) to 2 (mod 4). This means that

$$\delta w_2(c) = \beta v_2(c), \tag{3.11}$$

where $\beta$ is the Bockstein, which equals $Sq^1$ in this case. For even $n_c$, there is no such issue, and therefore

$$\delta w_2(c) = 0. \tag{3.12}$$

Combining, we can write

$$\delta w_2(c) = n_c \cdot \beta v_2(c). \tag{3.13}$$

When the right hand side is non-zero, $w_2(c)$ is not a cocycle anymore, and the coupling (3.7) is not well-defined. Indeed, the integrand is not closed:

$$\delta(Bw_2(c)) = n_c \cdot B\beta v_2(c) = n_c \cdot B\beta w_2(T), \qquad (3.14)$$

where we used (3.10). The rightmost expression depends only on the background fields, and therefore we can add the bulk $5d$ action

$$\exp\left(2\pi i \cdot \frac{1}{2}\int_{W_5} n_c \cdot B\beta w_2(T)\right) \qquad (3.15)$$

to make the combined bulk-boundary system non-anomalous. In other words, the gauge theory on the $4d$ boundary has a mixed 't Hooft anomaly between the $\mathbb{Z}_2$ 1-form symmetry and the spacetime rotation symmetry.

The existence of this mixed anomaly was first pointed out in [25], where $[SO(N_c)\times SU(N_f)]/\mathbb{Z}_2$ was used instead. In that paper it was also pointed out that this mixed anomaly between the magnetic $\mathbb{Z}_2$ 1-form symmetry and the rest of the symmetry in the $SO(2n_c)$ gauge theory is transformed into a 2-group structure between the electric $\mathbb{Z}_2$ 1-form symmetry and the rest of the symmetry in the $Spin(2n_c)$ gauge theory.

Let us briefly recall how this 2-group arises. The $Spin(2n_c)$ gauge theory has an electric $\mathbb{Z}_2$ 1-form symmetry, whose background field $E$ sets the Stiefel-Whitney class $w_2(c) \in H^2(X;\mathbb{Z}_2)$ of the $SO(2n_c)$ gauge bundle to be $E = w_2(c)$. Then, the relation (3.13) together with (3.10) results in

$$\delta E = n_c \cdot \beta w_2(T). \qquad (3.16)$$

This means that the electric $\mathbb{Z}_2$ 1-form symmetry extends the spacetime symmetry.[21] This extension can also be understood as the combination of the following two facts, namely that the $Spin(2n_c)$ theory is obtained by gauging $B$ of the $SO(2n_c)$ theory [24, 46], and that a theory with an anomaly (3.15) turns into a theory with an extension (3.16) under the gauging of the 1-form symmetry [44].

When we consider the $Spin$ QCD on non-spin manifolds in this manner, the orientation of the manifold alone does not suffice; we need an explicit cochain solving this equation and trivializing $\beta w_2(T) = w_3(T)$.[22] This is a higher analogue of the spin structure, which entails a choice of the trivialization of $w_2(T)$.

---

[21]When $n_c = 1$, the gauge group $Spin(2)$ is Abelian and the 1-form symmetry group is $U(1)$. In this case we can modify the background $E$ as $\tilde{E} = E - \frac{1}{2}w_2(T)$ since $E$ is now valued in $U(1)$, and the extension (3.16) can be resolved. In other words, the spacetime symmetry is not extended by the $U(1)$ 1-form symmetry, but the background $E$ that naturally extends to higher $n_c$ forms a sub-2-group of the direct product of the $U(1)$ 1-form and the spacetime symmetries. An Abelian gauge theory with an elaborated combination of matters can have a non-trivial 2-group global symmetry [47].

[22]In four dimensions this equation can always be solved, since the cohomology class $\beta w_2(T) = w_3(T)$ is known to vanish, which follows from the vanishing of $W_3(T)$, the integral Stiefel-Whitney class. We will give a proof below. In the discussion of the anomalies we also make use of five-manifolds. For them $w_3(T)$ does not necessarily vanish, with the Wu manifold as an example. Now let us provide a proof of the fact that $W_3(T) = 0$, or equivalently that $w_2(T)$ lifts to an integral class on all compact four-manifolds. We follow [48], which also gives a proof applicable to non-compact ones. Consider the universal coefficient sequences

$$\begin{array}{ccccccccc} 0 \to & \mathrm{Ext}_{\mathbb{Z}}(H_1(X;\mathbb{Z}),\mathbb{Z}) & \to H^2(X;\mathbb{Z}) \to & \mathrm{Hom}_{\mathbb{Z}}(H_2(X;\mathbb{Z}),\mathbb{Z}) & \to 0 \\ & a\downarrow & \downarrow & \downarrow b & \\ 0 \to & \mathrm{Ext}_{\mathbb{Z}}(H_1(X;\mathbb{Z}),\mathbb{Z}_2) & \to H^2(X;\mathbb{Z}_2) \to & \mathrm{Hom}_{\mathbb{Z}}(H_2(X;\mathbb{Z}),\mathbb{Z}_2) & \to 0 \end{array}.$$

To lift an element in $H^2(X;\mathbb{Z}_2)$, we need to lift elements in $\mathrm{Ext}_{\mathbb{Z}}(H_1(X;\mathbb{Z}),\mathbb{Z}_2)$ and $\mathrm{Hom}_{\mathbb{Z}}(H_2(X;\mathbb{Z}),\mathbb{Z}_2)$ via the maps $a$ and $b$. The map $a$ is clearly a surjection, since $\mathrm{Ext}_{\mathbb{Z}}(H_1(X;\mathbb{Z}),\mathbb{Z}) = \mathrm{Tors}\, H_1(X;\mathbb{Z})$ and $a$ is the mod 2 reduction. The remaining issue is to lift the map $w_2 : x \mapsto w_2(x) \in \mathbb{Z}_2$ for $x \in H_2(X;\mathbb{Z})$ from $\mathbb{Z}_2$-valued to $\mathbb{Z}$-valued. This is clearly possible if $w_2(x) = 0 \in \mathbb{Z}_2$ for torsion elements of $H_2(X;\mathbb{Z})$, and this last condition is indeed satisfied since $w_2(x)$ is the mod 2 reduction of the intersection product $x \cdot x \in \mathbb{Z}$.

# 4 Gauged WZW terms

Let us now discuss a few extra complications which arise when we introduce gauge fields into the discussions of the WZW terms. Before proceeding, we note that our approach here is to try to define and discuss the gauged WZW terms for the sigma model target space $X$ with an isometry group $G$ in general, without restricting to the case of 4d QCD unless necessary. An approach more directly focused toward the gauged WZW terms arising in 4d QCD can be found in [49, Sec. 3.1] and [40].

## 4.1 As differential forms

At the level of differential forms, the anomaly of a $d$-dimensional system with $G$ symmetry is given by its anomaly polynomial $\alpha(A, \omega)$, which is a gauge-invariant closed $(d + 2)$-form constructed from the background $G$-gauge field $A$ and the spin connection $\omega$ of the spacetime. Below we simply denote the anomaly by $\alpha(A)$ and leave the possible $\omega$ dependence implicit.

Let us now suppose that the symmetry $G$ spontaneously breaks to its subgroup $H$ in the infrared, resulting in the Nambu-Goldstone scalar field $\sigma : M_d \to G/H$. The crucial insight of [3] is that this process induces the WZW term for $\sigma$.

We assume that the Nambu-Goldstone field is the sole massless field in the infrared. Then, while the torsion part of the anomaly can be carried by other topological parts of the infrared theory, the sigma model part needs to reproduce the anomaly polynomial.[23] This can be achieved as follows.

The WZW term when $\sigma : M_d \to X$ extends to $\sigma : W_{d+1} \to X$ was given by the integral

$$e^{2\pi i \int_{W_{d+1}} \Gamma(\sigma)}. \tag{4.1}$$

We introduce the background gauge field $A$ for the flavor symmetry and generalize the coupling to

$$e^{2\pi i \int_{W_{d+1}} \underline{\Gamma}(\sigma, A)}, \tag{4.2}$$

where $\underline{\Gamma}(\sigma, A)$ is a possibly-non-closed but gauge-invariant $(d + 1)$-form such that

$$\Gamma(\sigma) := \underline{\Gamma}(\sigma, 0). \tag{4.3}$$

As we assumed that the sigma model field is the sole massless degree of freedom in the infrared, this coupling needs to reproduce the anomaly polynomial, meaning that it should have the same variation under the change of $W_{d+1}$ and the gauge field $A$ on it as the expression

$$e^{2\pi i \int_{W_{d+1}} \text{CS}(A)}, \tag{4.4}$$

where $\text{CS}(A)$ is the Chern-Simons term satisfying $\alpha(A) = d\text{CS}(A)$. This condition can be achieved by postulating

$$d\underline{\Gamma}(\sigma, A) = d\text{CS}(A) = \alpha(A). \tag{4.5}$$

Running the argument in reverse, this allows us to determine the ungauged WZW term starting from the anomaly. Namely, we solve (4.5) in terms of a not-necessarily-closed gauge-invariant differential form $\underline{\Gamma}(\sigma, A)$. We then set $A = 0$ to define $\Gamma(\sigma)$ as in (4.3). This process of obtaining the gauged and ungauged WZW term from the anomaly at the level of differential forms has been detailed in various sources. For example, the reader can find an explanation

---

[23]For $SU$ QCD with $N_c = 2$ and $N_f$ odd, the UV theory has the Witten's global $SU(2)$ anomaly [50]. Although this is a torsion part of the anomaly, this cannot be cancelled by gapped modes in the theory [51, Sec. 5] and has to be reproduced also by the sigma model part. More generally, any torsional anomaly not cancellable by gapped modes, recently discussed e.g. in [52, 53], needs to be reproduced by the sigma model.

of all the details in recent articles such as [54, Appendix C], [28] or [55], in which one can find not only the classic cases but also the cases with general $G$ and $H$.[24]

More mathematically, a gauge-invariant closed $(d+2)$-form $\alpha(A)$ constructed from the background $G$-gauge field $A$ determines an element $\alpha \in H^{d+2}(BG; \mathbb{R})$. Similarly, the closed $(d+1)$-form $\Gamma(\sigma)$ comes from an element $\Gamma \in H^{d+1}(G/H; \mathbb{R})$. The significance of the equation (4.5) is that $\alpha$ trivializes when pulled back to the total space of the universal $G/H$ bundle over $BG$. From the definition of the classifying spaces, this universal bundle is homotopy equivalent to $BH$, and we have the fibration

$$G/H \longrightarrow BH \xrightarrow{p} BG. \tag{4.6}$$

More generally, associated to a fibration

$$F \longrightarrow E \xrightarrow{p} B \tag{4.7}$$

we can consider the following operation. Take an element $\alpha \in H^{d+2}(B)$. Assume its pull-back to $E$ trivializes: $p^*(\alpha) = 0 \in H^{d+2}(E)$. At the cochain level, this means that there is an element $\underline{\Gamma} \in C^{d+1}(E)$ such that $\delta \underline{\Gamma} = p^*(\alpha)$. We now restrict $\Gamma$ to the fiber $F$ and write $\Gamma := \underline{\Gamma}|_F$. Then $\delta \Gamma = 0$, and therefore we have an element $\Gamma \in H^{d+1}(F)$. This operation is known as the *transgression* in algebraic topology, and $\Gamma$ is said to transgress to $\alpha$.[25] Summarizing, we find that, under the fibration (4.6),

- the gauged WZW term $\underline{\Gamma}$ is the non-closed cochain which trivializes the anomaly $\alpha$,

- the ungauged WZW term $\Gamma$ is the restriction of $\underline{\Gamma}$ which is closed and determines a cocycle,

- and the ungauged WZW term $\Gamma$ is said to transgress to the anomaly $\alpha$.

We note here that we made our analysis at the level of differential forms. This does not allow us, for example, to obtain the WZW term for $N_f = 2$ from the global anomaly of Witten for $SU(2)$ via transgression.[26] This was done in [18] using the differential version of the generalized cohomology theory $E^\bullet$, for $SU$ WZW terms. It would be useful to extend this technique to the differential version of the Anderson dual of the bordism group $(D\Omega_{\mathrm{spin}})^\bullet$, so that the transgression analysis can be performed also for $SO$ WZW terms for $N_f = 2$. An abstract version of the transgression map in the bordism case, constructing elements $\Gamma \in \mathrm{Inv}^d_{\mathrm{spin}}(F)$ from $\alpha \in \mathrm{Inv}^{d+1}_{\mathrm{spin}}(B)$ assuming that it trivializes in $\mathrm{Inv}^{d+1}_{\mathrm{spin}}(E)$, was also discussed in [56, Sec. 2.5].

## 4.2 Normalization of the ungauged WZW terms

Let us now check the normalization of the ungauged WZW terms for $SU$ and $SO$ QCD we discussed earlier. Here we perform this computation using algebraic topology, since the method using differential forms can be found elsewhere.

---

[24]The gauged WZW term in the original paper [20] had a few typos, where the gauged WZW term was constructed by trial and error. Systematic methods to obtain it were found slightly later in a number of independent papers, see e.g. [57–60].

[25]That the relation between the WZW term and the anomaly is the transgression is of course long known. See e.g. [61, Sec. 4], [62, Appendix] and [18, Sec. 5]. In particular, the reference [61, Sec. 4] already contains a detailed explanation of the transgression associated to the special but important case $G \to EG \to BG$.

[26]If we also consider the baryonic $U(1)$ symmetry and consider the QCD on a manifold with spin$^c$ structure, the global anomaly of Witten instead comes from the anomaly polynomial [63]. Then its transgression can be analyzed at the level of differential forms, and results in the ungauged WZW term (2.31) discussed in the latter part of Sec. 2.5.

For this purpose, we utilize the Leray-Serre spectral sequence (LSSS) associated to the fibration (4.7), which is a spectral sequence whose $E_2$ page is given by

$$E_2^{p,q} = H^p\big(B; H^q(F; \mathbb{Z})\big), \tag{4.8}$$

converging to $H^{p+q}(E; \mathbb{Z})$. The elements in $H^n(F; \mathbb{Z})$ which transgress to $H^{n+1}(B; \mathbb{Z})$ are known[27] to be those elements which survive to the $(n+1)$-st page $E_{n+1}^{0,n}$, and the transgression is known to coincide with the $(n+1)$-st differential

$$d_{n+1} : E_{n+1}^{0,n} \to E_{n+1}^{n+1,0}. \tag{4.9}$$

For the case at hand, we consider the fibration (4.6), and we know the cohomology groups of all three terms, as summarized in Appendix A. This allows us to determine the required transgression. For simplicity, we assume that $N_f$ is large enough so that the cohomology groups involved are in the generic range.

For $G = SU \times SU$ and $H = SU$, the LSSS is such that

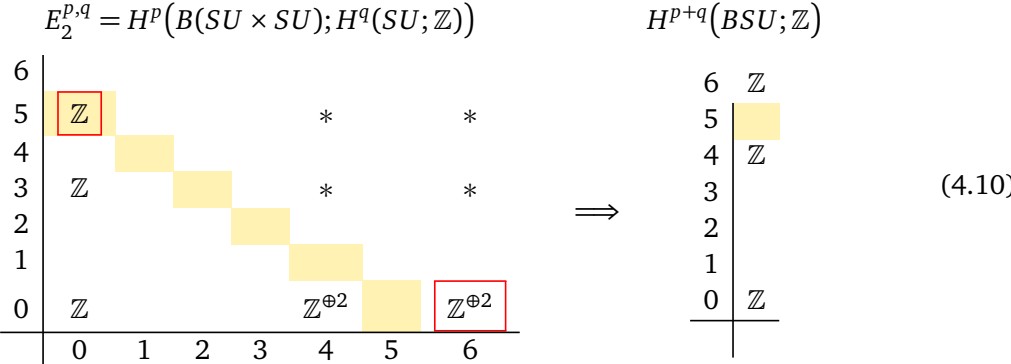

$$\tag{4.10}$$

To realize correct convergence in degree $p + q = 5, 6$, the differential $d_6 : E_6^{0,5} \to E_6^{6,0}$ should be an injection so that $E_7^{0,5} = E_\infty^{0,5} = 0$ and $E_7^{6,0} = E_\infty^{6,0} = \mathbb{Z}$. Considering the symmetry of exchanging two $SU$ factors, this means that the generator $x_5 \in H^5(SU; \mathbb{Z})$ transgresses to $c_3 - c_3' \in H^6(BSU \times BSU; \mathbb{Z})$. Since $\Gamma_5 = x_5/2$, the normalization (2.22) is verified.

Similarly, for $G = SU$ and $H = SO$, the LSSS must be such that

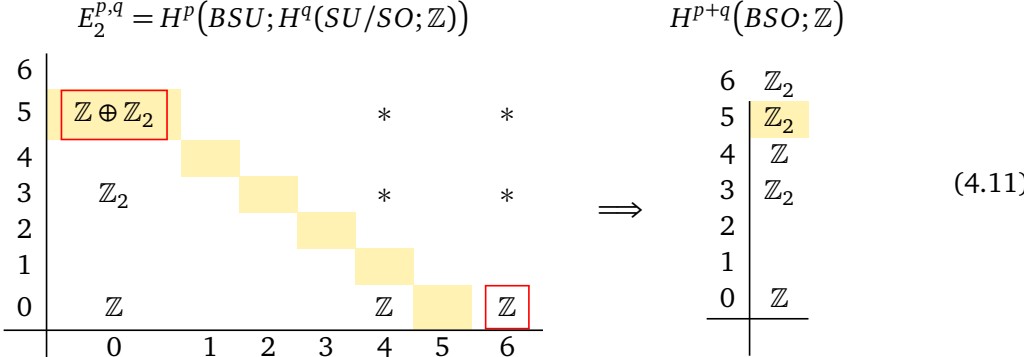

$$\tag{4.11}$$

To realize correct convergence in degree $p + q = 5, 6$, the differential $d_6 : E_6^{0,5} \to E_6^{6,0}$ should be a multiplication by 2 on the summand $\mathbb{Z}$ and the zero map on the summand $\mathbb{Z}_2$. Therefore the generator $y_5 \in H^5(SU/SO; \mathbb{Z})$ transgresses to $2c_3 \in H^6(SU; \mathbb{Z})$. Since $\Gamma_5 = y_5/2$, the normalization (2.35) is verified.

---

[27]For a gentle introduction to spectral sequences for physicists, see [64]. For the relation between the transgression and the Leray-Serre spectral sequence, see [65, Sec. 9.3].

### 4.3 Topological consistency of the gauged WZW term

#### 4.3.1 Generalities

Let us now consider possible global topological issues associated to the gauged WZW term (4.2). By construction, it has the same variation as the Chern-Simons term (4.4) under any small change in the data. This means that the combination

$$\exp\left(2\pi i \int_{W_{d+1}} \left(\mathrm{CS}(A) - \underline{\Gamma}(\sigma, A)\right)\right),\tag{4.12}$$

for closed manifolds $W_{d+1}$ determines a bordism invariant

$$\gamma : \Omega_{d+1}^{\mathrm{spin}}(BH) \to U(1),\tag{4.13}$$

where we again used the fact that the universal $G/H$ bundle over $BG$ is $BH$. We also note that by $\exp(2\pi i \int_{W_{d+1}} \mathrm{CS}(A))$ we mean the invertible phase describing the anomaly, not simply the differential form expression of the Chern-Simons term, which is not precise enough to discuss the torsion issues.

When (the torsion part of) $\gamma$ is non-zero, it signifies that the gauged WZW term itself is still anomalous. What we would like to do now is to determine $\gamma$. Note that (the torsion part of) $\gamma$ determines a $(d+1)$-dimensional spin invertible phase, and therefore gives an element of $\mathrm{Inv}_{\mathrm{spin}}^{d+1}(BH)$. Also, the deformation class of the expression (4.12) is the same as that of the Chern-Simons term alone, since $\underline{\Gamma}(\sigma, A)$ is a globally well-defined differential form. This means that $\gamma$ as an element of $\mathrm{Inv}_{\mathrm{spin}}^{d+1}(BH)$ is simply the pull-back of the original anomaly $\alpha \in \mathrm{Inv}_{\mathrm{spin}}^{d+1}(BG)$.

The preceding argument can be mathematically summarized in the following commutative diagram:

$$\begin{array}{ccccccccc}
0 \to & \mathrm{Ext}_{\mathbb{Z}}(\Omega_{d+1}^{\mathrm{spin}}(BH), \mathbb{Z}) & \xrightarrow{b} & \mathrm{Inv}_{\mathrm{spin}}^{d+1}(BH) & \xrightarrow{a} & \mathrm{Hom}_{\mathbb{Z}}(\Omega_{d+2}^{\mathrm{spin}}(BH), \mathbb{Z}) & \to 0 \\
& \uparrow & & \uparrow & & \uparrow & \\
0 \to & \mathrm{Ext}_{\mathbb{Z}}(\Omega_{d+1}^{\mathrm{spin}}(BG), \mathbb{Z}) & \xrightarrow{b} & \mathrm{Inv}_{\mathrm{spin}}^{d+1}(BG) & \xrightarrow{a} & \mathrm{Hom}_{\mathbb{Z}}(\Omega_{d+2}^{\mathrm{spin}}(BG), \mathbb{Z}) & \to 0.
\end{array}\tag{4.14}$$

Here, each row comes from the description of the invertible phases in terms of bordisms we recalled in (2.53), and the upward arrows are pull-backs along $p : BH \to BG$. We start from an anomaly in the lower middle part, $\alpha \in \mathrm{Inv}_{\mathrm{spin}}^{d+1}(BG)$. We pull it back to the upper middle part $\mathrm{Inv}_{\mathrm{spin}}^{d+1}(BH)$ and obtain $p^*(\alpha)$. We assumed that $a(p^*(\alpha))$ vanishes, i.e. the anomaly polynomial restricted to the unbroken subgroup $H$ is zero. This means that $p^*(\alpha)$ is in the image of $b$, so we can write $p^*(\alpha) = b(\gamma)$, where

$$\gamma \in \mathrm{Ext}_{\mathbb{Z}}(\Omega_{d+1}^{\mathrm{spin}}(BH), \mathbb{Z}) = \mathrm{Tors}\,\mathrm{Hom}(\Omega_{d+1}^{\mathrm{spin}}(BH), U(1)).\tag{4.15}$$

That $\gamma$ being non-zero signifies that there is a residual global anomaly in the gauged WZW term.

#### 4.3.2 Gauged WZW terms for QCD

The analysis so far was very general, and did not assume that the WZW term in question is actually the WZW term associated to QCD. In this concrete case, however, we can conclude that $\gamma$ is actually zero rather easily and there are no global topological issues. To see this, we note that in both cases

$$SU \to BSU \to B(SU \times SU)\tag{4.16}$$

and

$$SU/SO \to BSO \to BSU, \tag{4.17}$$

the middle term $BH$ is for the symmetry $H = SU$ or $SO$ under which the fermions in question can be given a non-zero mass. $\gamma \in \mathrm{Inv}^5_{\mathrm{spin}}(BH)$ is then the anomaly of fermions with respect to a symmetry $H$ under which they can be given a non-zero mass. This guarantees that not only the free part characterized by the anomaly polynomial vanishes, but also the subtler torsion part does so.

This quick argument, however, does not apply to the case when we consider $SO(2n_c)$ QCD on non-spin manifolds. In this case, the anomaly of the QCD takes values in $\mathrm{Inv}^5_{\mathrm{oriented}}(BSU)$. Its pull-back is in $\mathrm{Inv}^5_{\mathrm{oriented}}(BSO)$, which is not directly the anomaly of the fermions which can be made massive. Therefore we cannot argue that it vanishes, and indeed we will soon see that it is non-zero. So, let us continue the discussion of the general case and study how we can actually determine $\gamma$ in the non-zero case.

We make a simplifying assumption that the anomaly $\alpha$ comes from the cohomology class $\alpha \in H^{d+2}(BG; \mathbb{Z})$. In this case, instead of the commutative diagram (4.14), we can use the following:

$$
\begin{array}{ccccccccc}
0 & \to & \mathrm{Ext}_{\mathbb{Z}}(H_{d+1}(BH; \mathbb{Z}), \mathbb{Z}) & \xrightarrow{\beta} & H^{d+2}(BH; \mathbb{Z}) & \longrightarrow & \mathrm{Hom}_{\mathbb{Z}}(H_{d+2}(BH; \mathbb{Z}), \mathbb{Z}) & \to & 0 \\
 & & \uparrow & & \uparrow & & \uparrow & & \\
0 & \to & \mathrm{Ext}_{\mathbb{Z}}(H_{d+1}(BG; \mathbb{Z}), \mathbb{Z}) & \xrightarrow{\beta} & H^{d+2}(BG; \mathbb{Z}) & \longrightarrow & \mathrm{Hom}_{\mathbb{Z}}(H_{d+2}(BG; \mathbb{Z}), \mathbb{Z}) & \to & 0.
\end{array}
\tag{4.18}
$$

As recalled around (2.20), $\beta$ is simply the Bockstein homomorphism acting on

$$\mathrm{Ext}_{\mathbb{Z}}(H_{d+1}(BH; \mathbb{Z}), \mathbb{Z}) = \mathrm{Tors}\, H^{d+1}(BH, U(1)). \tag{4.19}$$

Therefore, determining $\gamma$ reduces to finding the element $\gamma \in \mathrm{Tors}\, H^{d+1}(BH, U(1))$ whose Bockstein $\beta(\gamma)$ equals the original anomaly $\alpha \in H^{d+2}(BG; \mathbb{Z})$ pulled back to $H^{d+2}(BH; \mathbb{Z})$.

Let us apply this consideration to the gauged WZW term for $SU$ QCD. Recall that the $SU$ WZW term is $N_c \Gamma_5 = (N_c/2)x_5$, where $x_5$ was a generator of $H^5(SU(N_f); \mathbb{Z})$, and that $x_5$ transgresses to $c_3 - c_3'$. Let us assume that $N_c = 2n_c$ is even, so that the anomaly $n_c(c_3 - c_3')$ is well-defined without the spin structure. In this case, as one can immediately derive from the universal coefficient theorem, $H^5(BSU(N_f); U(1)) = 0$. Therefore $\gamma$ is trivial, and the gauged WZW term is automatically well-defined.

Let us next study the gauged WZW term for $SO$ QCD. This time, the WZW term is $(N_c/4)y_5$, where $y_5$ is a generator of $\mathbb{Z} \subset H^5(SU(N_f)/SO(N_f); \mathbb{Z})$. Recall also that $y_5$ transgresses to $2c_3$. Then, consider the case when $N_c = 2n_c$ is even, so that the anomaly $n_c c_3$ is well-defined without the spin structure. Referring to the Appendix A, one can immediately read off the following:

- $c_3 \in H^6(BSU(N_f); \mathbb{Z})$ pulls back to $(W_3)^2 \in H^6(BSO(N_f); \mathbb{Z})$,

- which reduces to $(w_3)^2 \in H^6(BSO(N_f); \mathbb{Z}_2)$,

- which is the image of the Bockstein $\beta = Sq^1$ of $w_2 w_3 \in H^5(BSO(N_f); \mathbb{Z}_2)$.

This implies that the possible inconsistency of the gauged WZW term for the anomaly $\alpha = n_c c_3$ is given by

$$\gamma = n_c \cdot w_2(f) w_3(f), \tag{4.20}$$

where $w_i(f)$ is the Stiefel-Whitney classes of the $SO(N_f)$ bundle. This is non-zero and therefore the gauged WZW term is not well-defined at this point, on generic oriented manifolds.

This possible inconsistency disappears on spin manifolds, thanks to the following. We have $w_2 w_3 = w_2 Sq^1 w_2 = Sq^2 w_3$. Therefore,

$$\int_{W_5} w_2(f) w_3(f) = \int_{W_5} Sq^2 w_3(f) = \int_{W_5} v_2(T) w_3(f) \tag{4.21}$$

modulo 2, which is 0 mod 2 on a spin manifold as in (2.27). Therefore, the gauged WZW term is well-defined on a spin manifold, and this conclusion agrees with the general discussion we made at the beginning of this Sec. 4.3.2.

When $N_c$ is even, the $SO$ QCD can be put on non-spin manifolds, as we already mentioned in Sec. 3.3. In this case, the remaining inconsistency (4.20) disappears by a subtler mechanism. We will come back to this question in Sec. 5.2.2.

## 4.4 Torsion part of the gauged WZW term for $SO$ QCD

As the final topic in this section, we would like to discuss the issue of the torsion part of the $SO$ WZW term in the gauged case. Recall from our discussion in Sec. 2.6 that, for $N_f \geq 5$, the part $(N_c/4)y_5 = (N_c/2)\Gamma_5$ only specifies the ungauged $SO$ WZW term up to the addition of the torsion part specified by a character

$$\chi : \widetilde{\Omega}_4^{\text{spin}}(SU(N_f)/SO(N_f)) \to U(1). \tag{4.22}$$

To describe the gauged $SO$ WZW term, we need to describe its behavior for the sigma model fields taking values not only in $SU(N_f)/SO(N_f)$ but also in the total space $BSO(N_f)$ fibered over $BSU(N_f)$. As computed in Appendix B.2.1, we have the equality

$$\widetilde{\Omega}_4^{\text{spin}}(SU(N_f \geq 5)/SO(N_f)) \simeq \text{Tors}\, \widetilde{\Omega}_4^{\text{spin}}(BSO(N_f)) \simeq \mathbb{Z}_2. \tag{4.23}$$

This means that the unfixed torsion part of the gauged $SO$ WZW term simply comes from the character

$$\chi \in \mathbb{Z}_2 \subset \text{Hom}(\widetilde{\Omega}_4^{\text{spin}}(BSO(N_f)), U(1)). \tag{4.24}$$

Its non-trivial element is given by [45]

$$\exp\left(2\pi i \int_{M_4} \frac{1}{2}\mathcal{P}(w_2(f))\right), \tag{4.25}$$

where $w_2(f)$ is the second Stiefel-Whitney class of the $SO(N_f)$ bundle and $\mathcal{P} : H^2(X;\mathbb{Z}_2) \to H^4(X;\mathbb{Z}_4)$ is the Pontrjagin square. We also note that the pull-back of $w_2(f)$ to $SU(N_f)/SO(N_f)$ via $SU(N_f)/SO(N_f) \to BSO(N_f)$ is simply the generator of $H^2(SU(N_f)/SO(N_f);\mathbb{Z}_2) = \mathbb{Z}_2$. The analysis up to this point only determines the gauged WZW term for the $SO(N_c)$ QCD with $N_f$ flavors up to the addition of this torsion WZW term (4.25). It is at present difficult to specify exactly which, but see Appendix D.2.[28]

# 5 Solitonic symmetries

In this final section we would like to make some comments on the symmetries associated to the solitons in the low-energy non-linear sigma model.

---

[28]A proposal was recently made in [40, Sec. 4].

## 5.1  *SU* QCD

Let us first discuss the *SU* massless QCD, for which the target space of the low-energy sigma model is $SU(N_f)$. As $\pi_3(SU(N_f))$ and $H_3(SU(N_f))$ are naturally isomorphic by the Hurewicz theorem and are both equal to $\mathbb{Z}$, the low-energy sigma model has a single type of point-like solitons whose number as an integer is conserved. This quantum number has been identified as the baryon number [19, 20]. Let us recall why this should be the case.

As reviewed in Sec. 3.2, the baryon number and the $SU(N_f) \times SU(N_f)$ flavor symmetry has the anomaly characterized by the anomaly polynomial

$$N_c \left[ \frac{F}{2\pi} \right] (c_2 - c_2') \in H^6(B(U(1) \times SU(N_f) \times SU(N_f)); \mathbb{Z}), \tag{5.1}$$

where $c_2$ and $c_2'$ are the second Chern classes of the two $SU(N_f)$ factors, and $F = dA$ is the field strength of the background gauge field of the baryonic $U(1)$ symmetry.

We can inspect the LSSS (4.10) associated to the fibration

$$SU(N_f) \to BSU(N_f) \to B(SU(N_f) \times SU(N_f)), \tag{5.2}$$

and find that the generator $\Gamma_3 = x_3$ of $H^3(SU(N_f); \mathbb{Z}) = \mathbb{Z}$ transgresses to $c_2 - c_2'$. This means that the anomaly (5.1) transgresses from the WZW-type coupling

$$\exp\left( 2\pi i \cdot N_c \int_{W_5} \frac{F}{2\pi} \wedge \Gamma_3 \right), \tag{5.3}$$

which can be partially integrated to

$$\exp\left( i \cdot N_c \int_{M_4} A \wedge \Gamma_3 \right). \tag{5.4}$$

As $A$ is the background gauge field for the baryonic symmetry, $\Gamma_3$ should be the sigma-model expression of the baryonic number current [21–23].

We note that we already saw this coupling in Sec. 2.5 when we discussed how the *SU* WZW term for $N_f = 2$ can be described using differential forms. There, the term (5.4) is the sole WZW coupling of the system, without the $\exp(2\pi i \cdot N_c \int_{W_5} \Gamma_5)$ part.

## 5.2  *SO* QCD

### 5.2.1  Coupling to the 1-form $\mathbb{Z}_2$ symmetry

Let us next consider the massless QCD for the $\mathfrak{so}$ gauge algebra. The target space of the non-linear sigma model is $SU(N_f)/SO(N_f)$. For $Spin(N_c)$ QCD, this non-linear sigma model is the sole low-energy degree of freedom, but for $SO(N_c)_+$ QCD, there is an additional topological sector.

This can be inferred as follows. Let us recall that the $Spin(N_c)$ pure Yang-Mills is expected to confine with a single vacuum on any spatial manifold, while the $SO(N_c)$ pure Yang-Mills in the infrared is a $\mathbb{Z}_2$ gauge theory [24, 45, 66]. Now, the diagonal mass deformation $m\psi\psi + c.c.$ of the massless QCD becomes a potential term $m \operatorname{Tr} \sigma + c.c.$ in the sigma model field, and picks a unique vacuum as the lowest energy configuration.

This means that it is compatible to identify the low-energy limit of the $Spin(N_c)$ QCD as the sigma model with the target space $SU(N_f)/SO(N_f)$, as both of which result in a single vacuum after the mass deformation. Since the gauge group can be changed from $Spin(N_c)$ to $SO(N_c)$ by gauging the $\mathbb{Z}_2$ 1-form symmetry [24, 46], we see that the low-energy limit of the $SO(N_c)$

QCD is a $\mathbb{Z}_2$ gauge theory coupled to the sigma model with the target space $SU(N_f)/SO(N_f)$. Our final task is to find out how the two parts are coupled.

For this purpose, we study the soliton of the low-energy sigma model. In this case, the lowest non-trivial homotopy group is $\pi_2(SU(N_f)/SO(N_f)) = \mathbb{Z}_2$, which is then naturally isomorphic to $H_2(SU(N_f)/SO(N_f))$ via Hurewicz theorem. This gives rise to a string-like soliton whose tension is controlled by the dynamical scale of the QCD. Since the group is $\mathbb{Z}_2$, two copies of such a flux tube can annihilate together. This matches the property of the electric flux tube generated by a charge in the spinor representation of $Spin(N_c)$ in the confining phase [20]. If we employ a more modern terminology of $p$-form symmetries [24], this corresponds to the electric $\mathbb{Z}_2$ 1-form symmetry of $Spin(N_c)$ QCD, under which the Wilson lines in the spinor representation are charged. The charge operator is then the generator $w_2$ of $H^2(SU(N_f)/SO(N_f); \mathbb{Z}_2) = \mathbb{Z}_2$.

This allows us to determine the coupling of the $\mathbb{Z}_2$ gauge theory with the low-energy sigma model. As already mentioned, the $SO(N_c)$ gauge theory can be obtained from the $Spin(N_c)$ gauge theory by gauging its $\mathbb{Z}_2$ 1-form symmetry. This means that the low-energy limit of $SO(N_c)$ gauge theory has the coupling

$$\exp\left(\pi i \int_{M_4} (a\delta b + bw_2 + bB)\right). \tag{5.5}$$

Here, $a \in C^1(M_4; \mathbb{Z}_2)$ and $b \in C^2(M_4; \mathbb{Z}_2)$; $a\delta b$ is the kinetic term of the $\mathbb{Z}_2$ gauge theory; $w_2 \in H^2(M_4; \mathbb{Z}_2)$, which is the pull-back of the above mentioned class in $H^2(SU(N_f)/SO(N_f); \mathbb{Z}_2)$ by the sigma model field, measures the soliton charge; and we also included $B \in Z^2(M_4; \mathbb{Z}_2)$, which is the background for the magnetic $\mathbb{Z}_2$ 1-form symmetry of the $SO(N_c)$ gauge theory. We note that the equation of motion of (5.5) forces

$$B = w_2 \tag{5.6}$$

at the level of cohomology classes. In other words, the space of the sigma model field has disconnected components labelled by the cohomology class $w_2 \in H^2(M_4; \mathbb{Z}_2)$, and every sector other than the one specified by the background field $B$ through (5.6) is projected out from the path integral by the $\mathbb{Z}_2$ gauge theory. We also note that the torsion part (4.25) of the $SO$ WZW term we have not been able to fix, discussed in Sec. 4.4, is therefore essentially equal to the term

$$\exp\left(2\pi i \int_{M_4} \frac{1}{2}\mathcal{P}(B)\right). \tag{5.7}$$

The $SO$ QCD path integral should determine the action of the low-energy $\mathbb{Z}_2$ gauge theory plus the sigma model including the choice of this term, but at present we have not been able to pin it down.

### 5.2.2 On non-spin manifolds

In this final section we relax the constraint that our spacetime manifold is spin. This is possible when $N_c = 2n_c$ is even, as we saw in Sec. 3.3. When we consider $Spin(2n_c)$ gauge theory instead, we see that the electric $\mathbb{Z}_2$ 1-form symmetry extends the spacetime rotation group non-trivially as we saw in (3.16), which we reproduce here:

$$\delta E = n_c \cdot \beta w_2(T). \tag{5.8}$$

In the following we consider the more interesting case where $n_c$ is odd.

We do not study anomalies and invertible phases under this spacetime structure in any extensive manner here, but let us at least mention that the WZW terms are consistent. The ungauged WZW term was given in (2.35) and is given by

$$\exp\left(2\pi i \cdot N_c \int_{W_5} \frac{\Gamma_5}{2}\right) = \exp\left(2\pi i \cdot n_c \int_{W_5} \frac{y_5}{2}\right). \tag{5.9}$$

We therefore need to show that the integral of the generator $y_5$ of $H^5(SU/SO; \mathbb{Z})$ is even on a closed oriented 5-manifold with $E$ satisfying (5.8) is specified. Using the information gathered in Appendix A, this can be shown as follows.

The mod 2 reduction of $y_5$ is $w_2 w_3$, where $w_i \in H^i(SU/SO; \mathbb{Z}_2)$. Then we have

$$\int_{W_5} w_2 w_3 = \int_{W_5} Sq^2 w_3 = \int_{W_5} w_2(T) w_3 = \int_{W_5} w_2(T) \beta w_2 = \int_{W_5} w_2 \beta w_2(T), \quad (5.10)$$

where in the last equality we used the formula $\int a\beta b = \int b\beta a$ which follows because their sum is $\int \beta(ab) = \int w_1(T)ab = 0$. Now, our assumption (5.8) means that $\beta w_2(T)$ is cohomologically trivial, making $\int_{W_5} y_5 = \int_{W_5} w_2 w_3$ even.

If we use $SO(2n_c)$ instead of $Spin(2n_c)$ as the gauge group, the consistency is realized instead as follows. The equation of motion of the $\mathbb{Z}_2$ gauge theory (5.5) forces the magnetic 1-form symmetry background $B$ to equal the class $w_2$ of the WZW sigma model, as we saw above in (5.6). The integral of $y_5$ modulo 2 is still given by (5.10), which is now

$$= \int_{W_5} B\beta w_2(T). \tag{5.11}$$

This final expression only depends on the external background fields and not on the WZW sigma model fields which are path integrated. Therefore this expression gives the mixed anomaly of the system, and successfully reproduces the anomaly (3.15) of the $SO$ QCD.

The consistency of the gauged WZW term also follows in a similar manner. Indeed, the possible inconsistency is given by (4.20), which is also $w_2 w_3$, this time of the total space of the fibration $SU/SO \to BSO \to BSU$.

## Acknowledgements

The authors thank Nati Seiberg for suggesting them to study how to reproduce in the low-energy sigma model the mixed anomaly between $\mathbb{Z}_2$ 1-form symmetry and the flavor symmetry in $SO$ QCD. This suggestion eventually led to the analysis presented in this paper. They also thank Po-Shen Hsin and Ho Tat Lam for various illuminating discussions and for carefully reading the manuscript. The authors also thank Tetsu Nishimoto for the help in various computations in algebraic topology. In addition, the authors would like to thank J. P. Ang, Ben Gripaios, Sergei Gukov, Konstantinos Roumpedakis, Sahand Seifnashri, and Kazuya Yonekura for valuable comments on the v1 of the paper, which allowed the authors to improve the presentation of the manuscript greatly.

Y.L. is partially supported by the Programs for Leading Graduate Schools, MEXT, Japan, via the Leading Graduate Course for Frontiers of Mathematical Sciences and Physics and also by JSPS Research Fellowship for Young Scientists. Y.T. is partially supported by JSPS KAKENHI Grant-in-Aid (Wakate-A), No.17H04837 and JSPS KAKENHI Grant-in-Aid (Kiban-S), No.16H06335, and also by WPI Initiative, MEXT, Japan at IPMU, the University of Tokyo.

## A  Cohomology of $BSU$, $BSO$, $SU$ and $SU/SO$

In this appendix, we summarize the cohomology groups of the classifying spaces and the homogeneous spaces we need in this paper. We start with the following four well-known classical results, which can be found in many textbooks, including [67]:

$$
\begin{aligned}
H^*\big(BSU(n);\mathbb{Z}\big) &= \mathbb{Z}[c_2, c_3, \ldots, c_n], \\
H^*\big(BSO(n);\mathbb{Z}_2\big) &= \mathbb{Z}_2[w_2, \ldots, w_n], \\
H^*\big(SU(n);\mathbb{Z}\big) &= \textstyle\bigwedge_{\mathbb{Z}}[x_3, x_5, \ldots, x_{2n-1}], \\
H^*\big(SU(n)/SO(n);\mathbb{Z}_2\big) &= \textstyle\bigwedge_{\mathbb{Z}_2}[w_2, w_3, \ldots, w_n].
\end{aligned}
\tag{A.1}
$$

Here, $c_i$, $w_i$, $x_i$ has degree $2i$, $i$, $i$ respectively. We also note that the exterior algebra $\bigwedge[a_1, a_2, \ldots]$ is defined to be the polynomial algebra modulo the relations $a_1^2 = a_2^2 = \cdots = 0$ and $a_i a_j = -a_j a_i$; the former relation does not follow from the latter over $\mathbb{Z}_2$.

The $\mathbb{Z}_2$ cohomology of $BSU$ and $SU$ are simply mod 2 reductions of their integral counterparts:

$$
\begin{aligned}
H^*\big(BSU(n);\mathbb{Z}_2\big) &= \mathbb{Z}_2[c_2, c_3, \ldots, c_n], \\
H^*\big(SU(n);\mathbb{Z}_2\big) &= \textstyle\bigwedge_{\mathbb{Z}_2}[x_3, x_5, \ldots, x_{2n-1}].
\end{aligned}
\tag{A.2}
$$

The integral cohomology of $BSO$ is more involved [68, 69]; to degree 6 one has

| $d$ | 0 | 1 | 2 | 3 | 4 | 5 | 6 | $\cdots$ |
|---|---|---|---|---|---|---|---|---|
| $H^d(BSO(3);\mathbb{Z})$ | $\mathbb{Z}$ | 0 | 0 | $\mathbb{Z}_2$ | $\mathbb{Z}$ | 0 | $\mathbb{Z}_2$ | $\cdots$ |
| $H^d(BSO(4);\mathbb{Z})$ | $\mathbb{Z}$ | 0 | 0 | $\mathbb{Z}_2$ | $\mathbb{Z}^{\oplus 2}$ | 0 | $\mathbb{Z}_2$ | $\cdots$ |
| $H^d(BSO(n \geq 5);\mathbb{Z})$ | $\mathbb{Z}$ | 0 | 0 | $\mathbb{Z}_2$ | $\mathbb{Z}$ | $\mathbb{Z}_2$ | $\mathbb{Z}_2$ | $\cdots$ |
| generator | 1 | 0 | 0 | $W_3$ | $p_1$ ($e_4$) | $W_5$ | $W_3^2$ ($e_6$) | $\cdots$ |

(A.3)

where $W_i$ is an integral lift of $w_i$ and is of order 2, $p_1$ is the Pontrjagin class and reduces to $w_2^2$. We also note that there is an Euler class $e_n$ generating $\mathbb{Z}$ at degree $n$ when $n = 2k$ is even, which reduces to $w_n$.

The integral cohomology of $SU/SO$ is also complicated [70][29]:

| $d$ | 0 | 1 | 2 | 3 | 4 | 5 | 6 | 7 | $\cdots$ |
|---|---|---|---|---|---|---|---|---|---|
| $H^d(SU(3)/SO(3);\mathbb{Z})$ | $\mathbb{Z}$ | 0 | 0 | $\mathbb{Z}_2$ | 0 | $\mathbb{Z}$ | 0 | 0 | $\cdots$ |
| $H^d(SU(4)/SO(4);\mathbb{Z})$ | $\mathbb{Z}$ | 0 | 0 | $\mathbb{Z}_2$ | $\mathbb{Z}$ | $\mathbb{Z}$ | 0 | $\mathbb{Z}_2$ | $\cdots$ |
| $H^d(SU(5)/SO(5);\mathbb{Z})$ | $\mathbb{Z}$ | 0 | 0 | $\mathbb{Z}_2$ | 0 | $\mathbb{Z} \oplus \mathbb{Z}_2$ | 0 | $\mathbb{Z}_2$ | $\cdots$ |
| $H^d(SU(6)/SO(6);\mathbb{Z})$ | $\mathbb{Z}$ | 0 | 0 | $\mathbb{Z}_2$ | 0 | $\mathbb{Z} \oplus \mathbb{Z}_2$ | $\mathbb{Z}$ | $\mathbb{Z}_2$ | $\cdots$ |
| $H^d(SU(n \geq 7)/SO(n);\mathbb{Z})$ | $\mathbb{Z}$ | 0 | 0 | $\mathbb{Z}_2$ | 0 | $\mathbb{Z} \oplus \mathbb{Z}_2$ | 0 | $\mathbb{Z}_2^{\oplus 2}$ | $\cdots$ |
| generator | 1 | 0 | 0 | $W_3$ | $(e_4)$ | $y_5, W_5$ | $(e_6)$ | $a_7$ $W_7$ | $\cdots$ |

(A.4)

where $W_{2k+1}$ is the integral lift of $\beta w_{2k} = w_{2k+1}$, and $a_7$ is the integral lift of $\beta(w_2 w_4) = w_2 w_5 + w_3 w_4$ (therefore $a_7 = W_3 e_4$ for $n = 4$). Furthermore, $e_{2k}$ only exists when $n = 2k$ is even and reduces to $w_{2k}$, while $y_5$ generates $\mathbb{Z}$ and reduces to $w_2 w_3$.

---

[29]The authors thank Neil Strickland for the information. See https://mathoverflow.net/questions/345274/.

The relations between Chern classes and Stiefel-Whitney classes induced from the projection $\psi : BSO(n) \longrightarrow BSU(n)$ are

$$
\begin{array}{ccc}
H^*(BSU(n); \mathbb{Z}_2) & \xrightarrow{\psi^*} & H^*(BSO(n); \mathbb{Z}_2) \\
\rotatebox{90}{$\in$} & & \rotatebox{90}{$\in$} \\
c_i & \longmapsto & (w_i)^2
\end{array}
\tag{A.5}
$$

and

$$
\begin{array}{ccc}
H^*(BSU(n); \mathbb{Z}) & \xrightarrow{(-1)^i \cdot \psi^*} & H^*(BSO(n); \mathbb{Z}) \\
\rotatebox{90}{$\in$} & & \rotatebox{90}{$\in$} \\
c_{2i} & \longmapsto & p_i
\end{array}.
\tag{A.6}
$$

We also use the fact that the classes $w_i$ of $BSO(n)$ pull back to $w_i$ of $SU(n)/SO(n)$ along the map $\iota : SU(n)/SO(n) \to BSO(n)$; that is, we have

$$
\begin{array}{ccc}
H^*(BSO(n); \mathbb{Z}_2) & \xrightarrow{\iota^*} & H^*(SU(n)/SO(n); \mathbb{Z}_2) \\
\rotatebox{90}{$\in$} & & \rotatebox{90}{$\in$} \\
w_i & \longmapsto & w_i
\end{array}.
\tag{A.7}
$$

For $\mathbb{Z}_2$ cohomology, the actions of Steenrod squares $Sq^i$ are given by the Wu formulas

$$
Sq^{2i}(c_j) = \sum_{k=0}^{i} \binom{j-k-1}{i-k} c_{i+j-k} c_k \quad (0 \leq i \leq j),
\tag{A.8}
$$

$$
Sq^i(w_j) = \sum_{k=0}^{i} \binom{j-k-1}{i-k} w_{i+j-k} w_k \quad (0 \leq i \leq j),
\tag{A.9}
$$

and also from [32] one has

$$
Sq^{2i} x_{2j-1} = \binom{j-1}{i} x_{2i+2j-1}.
\tag{A.10}
$$

# B  Bordisms via Atiyah-Hirzebruch spectral sequence

In this appendix, we compute the spin bordism groups we need, namely those of $SU$, $SU/SO$, $BSU$ and $BSO$ via the Atiyah-Hirzebruch spectral sequence (AHSS), associated to the trivial fibration

$$
pt \longrightarrow X \xrightarrow{p} X.
\tag{B.1}
$$

We will not give an introduction to AHSS in this paper. An introduction for physicists can be found in [64]. Readable mathematical textbooks include [65].

In the following, we will freely use the bordism groups of $pt$ as known [71, 72]:

| $d$ | 0 | 1 | 2 | 3 | 4 | 5 | 6 | $\cdots$ | |
|---|---|---|---|---|---|---|---|---|---|
| $\Omega_d^{\mathrm{spin}}$ | $\mathbb{Z}$ | $\mathbb{Z}_2$ | $\mathbb{Z}_2$ | 0 | $\mathbb{Z}$ | 0 | 0 | $\cdots$ | (B.2) |
| $\Omega_d^{\mathrm{spin}^c}$ | $\mathbb{Z}$ | 0 | $\mathbb{Z}$ | 0 | $\mathbb{Z}^{\oplus 2}$ | 0 | $\mathbb{Z}^{\oplus 2}$ | $\cdots$ | |

We will also freely use the fact that our AHSS associated to the fibration (B.1) is such that the differentials going into the $p = 0$ column are all zero, see e.g. [65, below Theorem 9.10]. This follows from the splitting of $\Omega_d^{\mathrm{spin}}(X) = \Omega_d^{\mathrm{spin}}(pt) \oplus \widetilde{\Omega}_d^{\mathrm{spin}}(X)$ which we explained in (2.28).

Before proceeding, we note that for the spin bordism group of $SU/SO$, we provide an independent computation using the Adams spectral sequence in Appendix C.

## B.1  *SU*

### B.1.1  Spin bordism of *SU*

We first compute the spin bordism of $SU$, whose $E^2$ page is the following:

$$E^2_{p,q} = H_p\big(SU(N_f); \Omega^{\text{spin}}_q\big)$$

| | | | | | | | |
|---|---|---|---|---|---|---|---|
| 5 | | | | | | | |
| 4 | $\mathbb{Z}$ | | | $*$ | | $*$ | |
| 3 | | | | | | | |
| 2 | $\mathbb{Z}_2$ | | | $\mathbb{Z}_2$ | | $*$ | |
| 1 | $\mathbb{Z}_2$ | | | $\mathbb{Z}_2$ | | $\mathbb{Z}_2$ | |
| 0 | $\mathbb{Z}$ | | | $\mathbb{Z}$ | | $\mathbb{Z}$ | |
| | 0 | 1 | 2 | 3 | 4 | 5 | 6 |

(B.3)

Here, the columns to the right of the vertical dotted lines are to be discarded when $N_f = 2$.

The differential $\boxed{d_2 : E^2_{5,0} \to E^2_{3,1}}$ is known to be mod 2 reduction composed with the dual of $Sq^2$ [73], which turns out to be non-trivial for $N_f \geq 3$ since $Sq^2 x_3 = x_5$. Therefore, one obtains

$$\widetilde{\Omega}^{\text{spin}}_4(SU(N_f \geq 3)) = 0.$$

(B.4)

On the other hand, for $N_f = 2$ there is no $x_5$, and the $E^2$ page of the AHSS is given by that for $N_f \geq 3$ with the $p \geq 5$ part thrown away. As a result, non-trivial differentials do not exist and one is led to

$$\widetilde{\Omega}^{\text{spin}}_4(SU(2)) = \mathbb{Z}_2.$$

(B.5)

Furthermore, the differential $\boxed{d_2 : E^2_{5,1} \to E^2_{3,2}}$ is also known to be the dual of $Sq^2$ [73], and the $E^3$ page results in

| | | | | | | | |
|---|---|---|---|---|---|---|---|
| 6 | | | | | | | |
| 5 | | | | | | | |
| 4 | $\mathbb{Z}$ | | | $*$ | | $*$ | |
| 3 | | | | | | | |
| 2 | $\mathbb{Z}_2$ | | | | | $*$ | |
| 1 | $\mathbb{Z}_2$ | | | | | | |
| 0 | $\mathbb{Z}$ | | | $\mathbb{Z}$ | | $\mathbb{Z}$ | |
| | 0 | 1 | 2 | 3 | 4 | 5 | 6 |

(B.6)

From this, one finds out that

$$\Omega^{\text{spin}}_5(SU(N_f \geq 3)) = \mathbb{Z}$$

(B.7)

and

$$\Omega^{\text{spin}}_6(SU(N_f \geq 3)) = 0.$$

(B.8)

Note that the map $\Omega^{\text{spin}}_5(SU(N_f)) = \mathbb{Z} \to H_5(SU(N_f); \mathbb{Z}) = \mathbb{Z}$ is a multiplication by two, since $E^\infty_{5,0} = E^3_{5,0} = \text{Ker}\,\boxed{d_2 : E^2_{5,0} \to E^2_{3,1}} = 2\mathbb{Z}$.

### B.1.2 Spin$^c$ bordism of $SU$

Similarly, one can compute the spin$^c$ bordism of $SU$, starting from the following $E^2$ page:

$$E^2_{p,q} = H_p\big(SU(N_f); \Omega_q^{\text{spin}^c}\big)$$

(B.9)

Since degree $p + q = 4$ entries (except $E^2_{0,4}$) are empty, one immediately obtains

$$\widetilde{\Omega}_4^{\text{spin}^c}(SU(N_f)) = 0,$$

(B.10)

independent of whether $N_f = 2$ or $N_f \geq 3$. This means that the WZW term for $N_f = 2$ with spin$^c$ structure is not of the torsion type (as was in the spin structure case) but of the free type.

### B.1.3 WZW terms for $SU$ QCD from cobordism

Let us comment on how the discrete WZW term for $N_f = 2$ is related to the ordinary WZW term for $N_f \geq 3$, from the cobordism point of view. We note that the WZW term is an invertible phase associated to $SU(N_f)$. According to [12] and as we reviewed in the main part of the paper, fermionic invertible phases in dimension $d$ associated to $X$, up to continuous deformations, can be classified by $\text{Inv}^d_{\text{spin}}(X)$, which is the Anderson dual of the spin bordism, shifted by $-1$. Therefore it can be directly computed by the AHSS, without computing the bordism groups $\Omega_d^{\text{spin}}(X)$ first. For $X = SU(N_f)$, the $E_2$ page is given by

$$E^{p,q}_2 = H^p\big(SU(N_f); \text{Inv}^q_{\text{spin}}\big)$$

(B.11)

This $E_2$ page is compatible with our discussion above, showing

$$0 \to \underbrace{H^5(SU(N_f); \mathbb{Z})}_{E_2^{5,-1} = \mathbb{Z}} \to \underbrace{\text{Inv}^4_{\text{spin}}(SU(N_f))}_{\mathbb{Z}} \to \underbrace{\text{Inv}^4_{\text{spin}}(SU(2))}_{E_2^{3,1} = \mathbb{Z}_2} \to 0.$$

(B.12)

This reads: the 4$d$ WZW term for general $SU(N_f)$ on *oriented* manifolds is given by $H^5(SU(N_f); \mathbb{Z})$. This maps to twice the minimal WZW term for general $SU(N_f)$ on *spin* manifolds, and their difference comes from the discrete WZW term for $SU(2)$ on spin manifolds.

### B.2    $SU/SO$

#### B.2.1    Spin bordism of $SU/SO$

We can obtain the spin bordism of $SU/SO$ in a similar manner. When $N_f = 2$ we have $SU(2)/SO(2) = S^2$, and therefore $\widetilde{\Omega}_d^{\text{spin}}(SU(2)/SO(2)) = \Omega_{d-2}^{\text{spin}}(pt)$ from the suspension isomorphism. The non-trivial cases are then $N_f \geq 3$. The $E^2$ pages are now as follows:

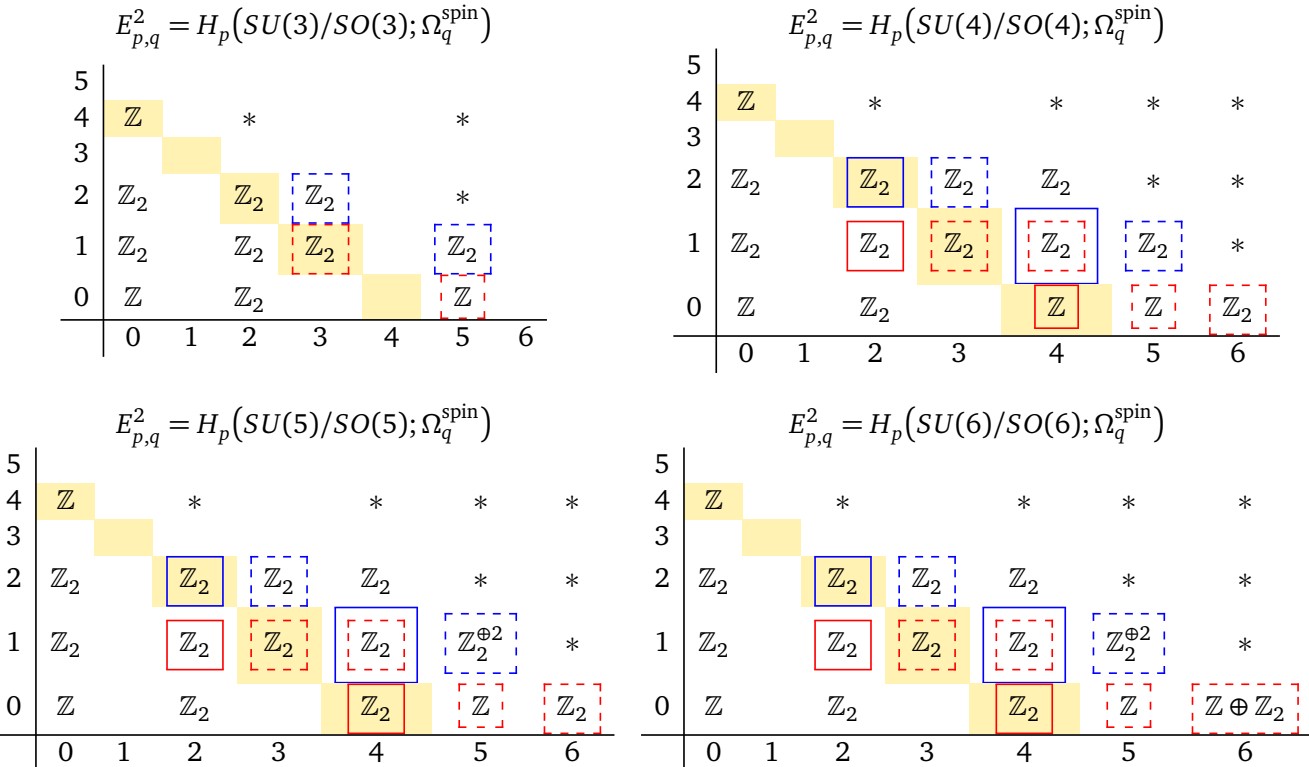

(B.13)

The differentials $\boxed{d_2 : E_{4,0}^2 \to E_{2,1}^2}$, $\dashbox{d_2 : E_{5,0}^2 \to E_{3,1}^2}$ and $\dashbox{d_2 : E_{6,0}^2 \to E_{4,1}^2}$ are again given by the mod 2 reduction composed with the dual of $Sq^2$. The first one turns out to be zero since $Sq^2 w_2 = (w_2)^2 = 0$, while the remaining two are possibly non-trivial since $Sq^2 w_3 = w_5 + w_3 w_2$ and $Sq^2 w_4 = w_6 + w_4 w_2$. The other differentials $\boxed{d_2 : E_{4,1}^2 \to E_{2,2}^2}$ and $\dashbox{d_2 : E_{5,1}^2 \to E_{3,2}^2}$ are again the dual of $Sq^2$, and the same argument tells that the former is always trivial while the

latter can be non-trivial. Therefore, we arrive at the $E^3$ pages given as follows:

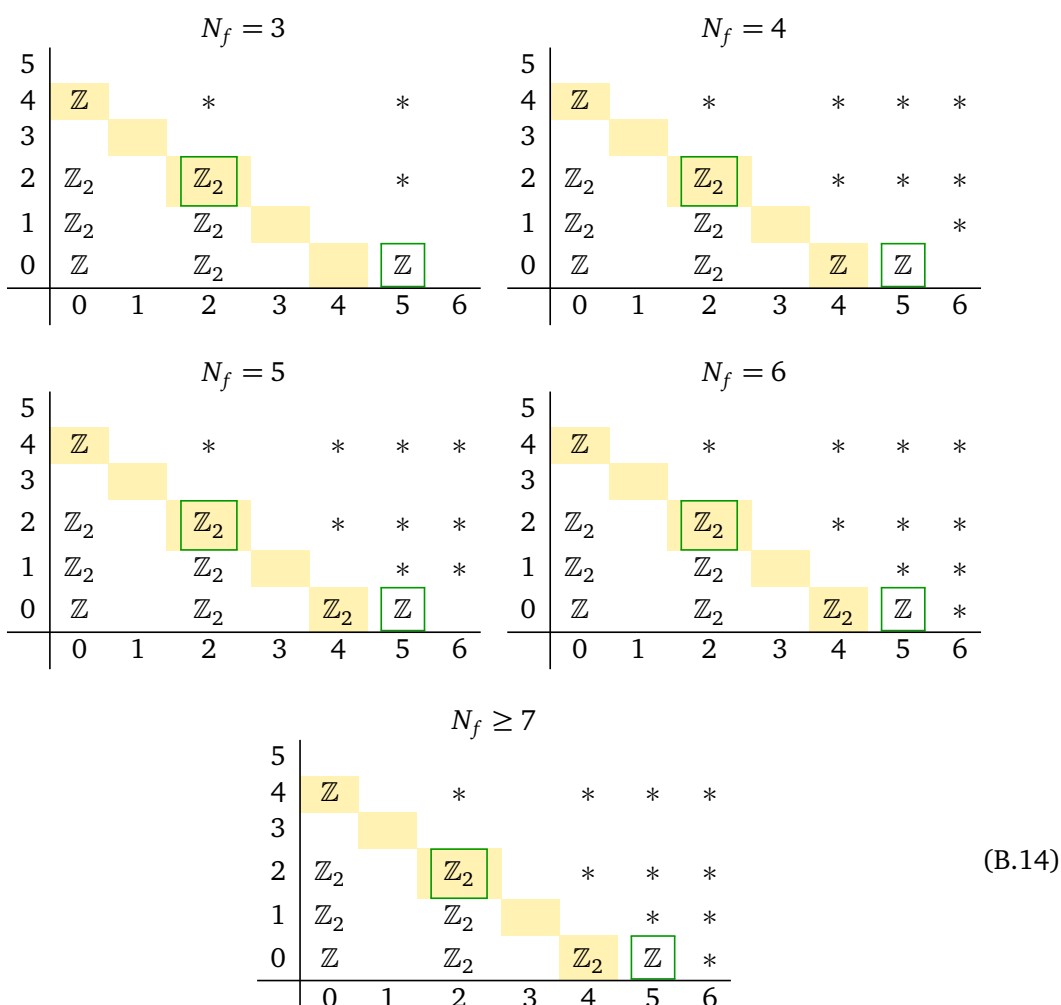

$$\tag{B.14}$$

It further turns out that $\boxed{d_3 : E^3_{5,0} \to E^3_{2,2}}$ is also non-trivial. Indeed, if this is trivial, then the summand $\mathbb{Z}$ in $\Omega^{\text{spin}}_5(SU(N_f)/SO(N_f))$ is $2\mathbb{Z} \subset \mathbb{Z} = H_5(SU(N_f)/SO(N_f); \mathbb{Z})$, which contradicts with the fact that $y_5$, the generator of $\mathbb{Z} = H^5(SU(N_f)/SO(N_f); \mathbb{Z})$, always integrates to multiples of four on spin manifolds, as we will see in Appendix C and Appendix D.1. It should also be possible to check the non-triviality of $d_3$ using the general form of $d_3$ in terms of cochains determined in [74].

As a result, one is led to

$$\widetilde{\Omega}^{\text{spin}}_4(SU(N_f)/SO(N_f)) = \begin{cases} \mathbb{Z}_2 & (N_f \geq 5) \\ \mathbb{Z} & (N_f = 4) \\ 0 & (N_f = 3) \end{cases}, \tag{B.15}$$

and

$$\Omega^{\text{spin}}_5(SU(N_f)/SO(N_f)) = \mathbb{Z}. \tag{B.16}$$

Before proceeding, we would like to stress again that

$$\underbrace{\Omega^{\text{spin}}_5(SU(N_f)/SO(N_f))}_{=\mathbb{Z}} \to \underbrace{H_5(SU(N_f)/SO(N_f))}_{=\mathbb{Z}} \tag{B.17}$$

is a multiplication by four. We also would like to note that, due to the naturality of the AHSS, the map

$$\underbrace{\widetilde{\Omega}_4^{\mathrm{spin}}(SU(4)/SO(4))}_{=\mathbb{Z}} \to \underbrace{\widetilde{\Omega}_4^{\mathrm{spin}}(SU(5)/SO(5))}_{=\mathbb{Z}_2}, \qquad (B.18)$$

comes from

$$\underbrace{\tilde{H}_4(SU(4)/SO(4);\mathbb{Z})}_{=\mathbb{Z}} \to \underbrace{\tilde{H}_4(SU(5)/SO(5);\mathbb{Z})}_{=\mathbb{Z}_2}, \qquad (B.19)$$

which is the mod 2 reduction.

### B.2.2 WZW terms for $SO$ QCD from cobordism

As in Sec. B.1.3, let us see how the discrete WZW term for $N_f = 2$ is related to the ordinary WZW term for $N_f \geq 3$, from the cobordism point of view. The $E_2$ page of the relevant AHSS is as follows:

$$E_2^{p,q} = H^p\big(SU(N_f)/SO(N_f);\mathrm{Inv}_{\mathrm{spin}}^q\big)$$

| q \ p | 0 | 1 | 2 | 3 | 4 | 5 | 6 |
|---|---|---|---|---|---|---|---|
| 4 | | | | | | | |
| 3 | $\mathbb{Z}$ | | | $*$ | | $*$ | $*$ |
| 2 | $\mathbb{Z}_2$ | | $\mathbb{Z}_2$ | $*$ | $*$ | $*$ | $*$ |
| 1 | $\mathbb{Z}_2$ | | $\mathbb{Z}_2$ | $\mathbb{Z}_2$ | $*$ | $*$ | $*$ |
| 0 | | | | | | | |
| −1 | $\mathbb{Z}$ | | | $\mathbb{Z}_2$ | $(\mathbb{Z})$ | $\mathbb{Z}(\oplus\mathbb{Z}_2)$ | $*$ |

$$(B.20)$$

As we discussed repeatedly, the free part of the 4$d$ WZW term for general $SU(N_f)/SO(N_f)$ on *oriented* manifolds which is given by $H^5(SU(N_f)/SO(N_f);\mathbb{Z})$, maps to four times the (free part of the) minimal WZW term for general $SU(N_f)/SO(N_f)$ on *spin* manifolds. The $E_2$ page above says that this factor of four arises due to the extension of the direct summand $\mathbb{Z}$ in $E_{5,-1}^2$ once by $\mathbb{Z}_2 = E_2^{3,1}$ and then again by $\mathbb{Z}_2 = E_2^{2,2}$. This last $\mathbb{Z}_2$ is already present when $N_f = 2$, meaning that the generator of the direct summand $\mathbb{Z}$ of $\mathrm{Inv}_{\mathrm{spin}}^4(SU(N_f)/SO(N_f))$ pulls back to the generator of $\mathbb{Z}_2$ of $\mathrm{Inv}_{\mathrm{spin}}^4(SU(2)/SO(2))$.

### B.3 $BSU$

We can keep going on and calculate the spin bordism of $BSU$. Starting from the $E^2$ page

$$E_{p,q}^2 = H_p\big(BSU(N_f);\Omega_q^{\mathrm{spin}}\big)$$

| q \ p | 0 | 1 | 2 | 3 | 4 | 5 | 6 | 7 |
|---|---|---|---|---|---|---|---|---|
| 6 | | | | | | | | |
| 5 | | | | | | | | |
| 4 | $\mathbb{Z}$ | | | | $*$ | | $*$ | |
| 3 | | | | | | | | |
| 2 | $\mathbb{Z}_2$ | | | | $\mathbb{Z}_2$ | | $*$ | |
| 1 | $\mathbb{Z}_2$ | | | | $\mathbb{Z}_2$ | | $\mathbb{Z}_2$ | |
| 0 | $\mathbb{Z}$ | | | | $\mathbb{Z}$ | | $\mathbb{Z}$ | |

$$(B.21)$$

and proceeding with the same logic using $Sq^2 c_2 = c_3$ modulo 2 from the Wu formula, one obtains

$$\Omega_5^{\mathrm{spin}}(BSU(N_f)) = 0 \qquad (B.22)$$

and

$$\Omega_6^{\mathrm{spin}}(BSU(N_f)) = \mathbb{Z}. \tag{B.23}$$

Again, the map $\Omega_6^{\mathrm{spin}}(BSU(N_f)) = \mathbb{Z} \to H_6(BSU(N_f); \mathbb{Z}) = \mathbb{Z}$ is a multiplication by two.

### B.4 *BSO*

Similarly for the spin bordism of *BSO*, we have the following $E^2$ pages:

$$E_{p,q}^2 = H_p\big(BSO(3); \Omega_q^{\mathrm{spin}}\big)$$

$$E_{p,q}^2 = H_p\big(BSO(4); \Omega_q^{\mathrm{spin}}\big)$$

$$E_{p,q}^2 = H_p\big(BSO(N_f \geq 5); \Omega_q^{\mathrm{spin}}\big) \tag{B.24}$$

Since $Sq^2 w_2 = (w_2)^2$, $Sq^2 w_3 = w_5 + w_3 w_2$, $Sq^2(w_2)^2 = (w_3)^2$, and $Sq^2 w_4 = w_6 + w_4 w_2$, the differentials marked above are all non-trivial. Resulting $E^3$ pages are given as follows:

$$N_f = 3 \qquad\qquad N_f = 4$$

$$N_f \geq 5 \tag{B.25}$$

As a result, one can read off

$$
\widetilde{\Omega}_4^{\mathrm{spin}}(BSO(N_f)) = \begin{cases} \mathbb{Z} \oplus \mathbb{Z}_2 & (N_f \geq 5) \\ \mathbb{Z}^{\oplus 2} & (N_f = 4) \\ \mathbb{Z} & (N_f = 3) \end{cases} \tag{B.26}
$$

and

$$
\Omega_5^{\mathrm{spin}}(BSO(N_f)) = 0. \tag{B.27}
$$

We note that the result (B.26) corresponds to the fact that $4d$ $SO$ gauge theories have the standard theta angle (for the $\mathbb{Z}$ summand) and a discrete theta angle (for the $\mathbb{Z}_2$ summand) [45]. Using characteristic classes, this comes from the fact that[30]

$$
p_1 \equiv 2w_4 + \mathcal{P}(w_2) \pmod 4, \tag{B.28}
$$

where $\mathcal{P}$ is the Pontrjagin square. On a spin manifold $\mathcal{P}(w_2)$ is even, and we have

$$
\frac{p_1}{2} \equiv w_4 + \frac{1}{2}\mathcal{P}(w_2) \pmod 2. \tag{B.29}
$$

The left hand side is the dual of the generator of the $\mathbb{Z}$ summand, and either of the terms of the right hand side can be taken to be the dual of the generator of the $\mathbb{Z}_2$ summand.

Recalling that the classes $w_i$ of $SU/SO$ are the pull-backs of the classes $w_i$ of $BSO$ and comparing the spectral sequences, we find that the dual of the generator of $\mathbb{Z}_2 = \widetilde{\Omega}_4^{\mathrm{spin}}(SU/SO)$ can be taken to be $w_4 = \mathcal{P}(w_2)/2$.

**Absence of mixed anomalies:** Here we examine the mixed anomaly between $SO(N_c)$ and $SU(N_f)$ for fermion systems charged under $G' = SO(N_c) \times SU(N_f)$, from the cobordism point of view. This assures the naive intuition that the flavor symmetry of $4d$ $SO$ QCD is $SU$. The $E_2$ page of the relevant AHSS can be filled by applying the Künneth formula[31] to the data in Appendix A, and is given as follows:

$$
E_2^{p,q} = H^p\big(BSO \times BSU; \mathrm{Inv}_{\mathrm{spin}}^q\big)
$$

| | 0 | 1 | 2 | 3 | 4 | 5 | 6 |
|---|---|---|---|---|---|---|---|
| 4 | | | | | | | |
| 3 | $\mathbb{Z}$ | | | $*$ | $*$ | $*$ | $*$ |
| 2 | $\mathbb{Z}_2$ | | $\mathbb{Z}_2$ | $*$ | $*$ | $*$ | $*$ |
| 1 | $\mathbb{Z}_2$ | | $\mathbb{Z}_2$ | $\mathbb{Z}_2$ | $*$ | $*$ | $*$ |
| 0 | | | | | | | |
| −1 | $\mathbb{Z}$ | | | $\mathbb{Z}_2$ | $\mathbb{Z}^{\oplus 2}$ | $\mathbb{Z}_2$ | $*$ |

$$\tag{B.30}$$

Fortunately, since the non-trivial mixing between $H^*(BSO;\mathbb{Z})$ and $H^*(BSU;\mathbb{Z})$ occurs above degree 7 (similarly above degree 6 for $\mathbb{Z}_2$-cohomology), elements of $\mathrm{Inv}_{\mathrm{spin}}^{p+q\leq 5}(BSO \times BSU)$ should be exhausted by the pull-backs from those of $\mathrm{Inv}_{\mathrm{spin}}^{d\leq 5}(BSO)$ and $\mathrm{Inv}_{\mathrm{spin}}^{d\leq 5}(BSU)$. This means that there is no mixed anomaly between $SO$ and $SU$ at least for spacetime dimensions

---

[30]The proof can be found in [75]. See also https://mathoverflow.net/questions/166280/.

[31]When the coefficient is a field $F$ it is simply $H^d(X \times Y; F) = \bigoplus_{p+q=d} H^p(X;F) \otimes H^q(Y;F)$. For the $\mathbb{Z}$-coefficient case it is given by

$$
0 \to \bigoplus_{p+q=d} H^p(X;\mathbb{Z}) \otimes H^q(Y;\mathbb{Z}) \to H^d(X \times Y;\mathbb{Z}) \to \bigoplus_{p+q=d+1} \mathrm{Tor}_{\mathbb{Z}}(H^p(X;\mathbb{Z}), H^q(Y;\mathbb{Z})) \to 0,
$$

which is known to split non-canonically. See e.g. [76, Theorem 5.5.11]. In our case, $H^{\bullet}(BSU;\mathbb{Z})$ is free, so the Tor term vanishes.

$\leq 4$. Also, note that since there is no interference between cohomologies of $BSU$ and $BSO$, one can stack two AHSS (B.21) and (B.24) for the region $p + q \leq 5$ and deduce

$$\Omega_d^{\text{spin}}(BSO \times BSU) = \Omega_d^{\text{spin}}(BSO) \oplus \Omega_d^{\text{spin}}(BSU) \quad (d \leq 5). \tag{B.31}$$

Furthermore, a similar argument applies to the $G' = [Spin(\text{spacetime}) \times SO(2n_c)]/\mathbb{Z}_2 \times SU(2n_f)$ case. The relevant bordism in this case is *twisted* spin bordism $\text{Inv}_{[Spin \times SO]/\mathbb{Z}_2}^d(BSU)$, where the $E_2$ page of the AHSS converging to it is given by $E_2^{p,q} = H^p(B(SO/\mathbb{Z}_2) \times BSU; \text{Inv}_{\text{spin}}^d)$. From the LSSS associated to the fibration $BSO \to B(SO/\mathbb{Z}_2) \to K(\mathbb{Z}_2, 2)$ (which we omit the detail), one finds that the lowest degree of non-trivial elements in $H^*(B(SO(2n_c)/\mathbb{Z}_2); \mathbb{Z})$ is 3 (and accordingly that in $H^*(B(SO(2n_c)/\mathbb{Z}_2); \mathbb{Z}_2)$ is 2). Therefore, there is no non-trivial mixing between $B(SO/\mathbb{Z}_2)$ and $BSU$ at $p + q \leq 5$, as in the previous case.

# C Bordisms via Adams spectral sequence

In this appendix we provide an independent computation of $\Omega_\bullet^{\text{spin}}(SU/SO)$, which plays an important role in this paper, using the Adams spectral sequence.

The Adams spectral sequence (ASS) and the Atiyah-Hirzebruch spectral sequence (AHSS) have complementary features. In the authors' opinion, ASS has the following merits over AHSS:

- ASS tends to be more powerful than AHSS.

- Some extensions in the $E_\infty$ page can be made manifest in ASS.

while it has the following demerits:

- The $E_2$ page of ASS is harder to compute, since it requires much more homological algebra.

- The physical meaning of ASS is unclear at present, while the AHSS can be physically interpreted as decorating the domain walls and their intersections by invertible phases [13, 77].

A readable account of ASS can be found in [78], which was written by mathematicians for those interested in the (co)bordism classification of invertible phases.

The ASS, specialized to the cases we are after, is

$$\text{Ext}_{A(1)}^{s,t}(H^*(X; \mathbb{Z}_2), \mathbb{Z}_2) \Rightarrow \text{ko}_{t-s}(X)_2^\wedge, \tag{C.1}$$

where $A(1)$ is the algebra generated by the Steendrod operations $Sq^1$ and $Sq^2$, depicted in Fig 2, $\text{Ext}_{A(1)}$ is the Ext functor in the category of $\mathbb{Z}$-graded $A(1)$ modules, $t$ is the $\mathbb{Z}$-grading on the modules, $s$ is the degree of the Ext itself. $\text{ko}_\bullet$ is the connected $KO$ homology, which agrees with the spin-bordism when the degree $\leq 7$. $_2^\wedge$ means the 2-completion. In practice, the 2-completion removes $k$-torsion parts for odd $k$, and replaces the free part $\mathbb{Z}$ by the module of 2-adic integers $\mathbb{Z}_2^\wedge$. Because we are not interested in the direct summand $\text{ko}_\bullet(pt)$, we use the reduced version:

$$\text{Ext}_{A(1)}^{s,t}(\tilde{H}^*(X; \mathbb{Z}_2), \mathbb{Z}_2) \Rightarrow \widetilde{\text{ko}}_{t-s}(X)_2^\wedge. \tag{C.2}$$

We will now examine $X = SU(N_f)/SO(N_f)$ first in the simplest non-trivial case $N_f = 3$, and then the generic case $N_f \geq 7$, and then the intermediate cases $N_f = 6, 5, 4$ in this order.

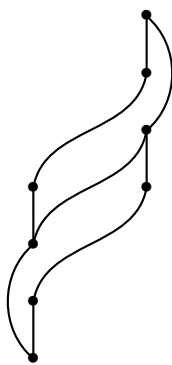

Figure 2: The algebra $A(1)$, which is generated by $Sq^1$ and $Sq^2$. Each dot represents a basis over $\mathbb{F}_2$, whose vertical position expresses its degree. Each vertical straight line represents the action $Sq^1$ and other curved lines are the $Sq^2$ action.

$N_f = 3$ : First let us compute the $E_2$ page for $X = SU(3)/SO(3)$. The $A(1)$ module structure of $M = \tilde{H}^*(X; \mathbb{Z}_2)$ is

$$
\begin{array}{l}
\bullet\, w_2 w_3 \\
\\
\bullet\, w_3 \\
\bullet\, w_2
\end{array}
\tag{C.3}
$$

To compute $\text{Ext}(M, \mathbb{Z}_2)$, we compute the minimal free (projective) resolution

$$
0 \leftarrow M \leftarrow P_0 \leftarrow P_1 \leftarrow \cdots , \tag{C.4}
$$

where each module $P_i$ are free up to grade-shift $[t]$:

$$
P_s = \bigoplus_{j=1}^{N_s} A(1)[t_{s,j}], \tag{C.5}
$$

with some integers $N_s$ and $t_{s,j}$. We take the resolution to be minimal, i.e. we take $P_0$ so that $N_0$ to be minimal, then take $P_1$ so that $N_1$ to be minimal without changing the already determined $P_0$, and so forth. The general theorem on Hopf algebra asserts that the morphisms between $P_i$ become trivial after passed to the functor $\text{Hom}(-, \mathbb{Z}_2)$ and thus

$$
\text{Ext}_{A(1)}^{*,*}(M, \mathbb{Z}_2) = \bigoplus_s \bigoplus_j^{N_s} \mathbb{Z}_2[s, t_{s,j}], \tag{C.6}
$$

where $[s, t]$ is the bidegree shift. Once the $E_2$ page is determined, it is convenient to draw a chart, called an *Adams chart*, in which for each $(s, j)$ a dot is drawn at $(t_{s,j} - s, s)$ in the $(t - s, s)$ coordinate.

For the particular $M$ (C.3), noting that the bottom element $w_2$ has $(t$-)degree 2, we find

the following exact sequence

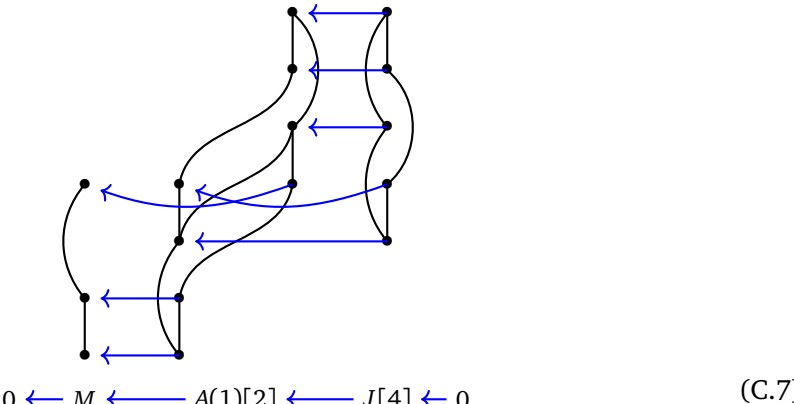

$$0 \longleftarrow M \longleftarrow A(1)[2] \longleftarrow J[4] \longleftarrow 0 \qquad \text{(C.7)}$$

In particular, we get $P_0 = A(1)[2]$. This amounts to a dot at $(t-s,s) = (2,0)$ in the Adams chart. Although one can keep going to resolve the $J$ ("Joker"), its Adams chart is given in [78, Fig. 29], so we can just migrate it up to a shift, resulting in the following Adams chart for $M$, where the vertical axis is $s$ and the horizontal axis is $t-s$:

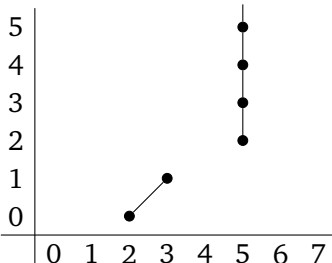

Fortunately, this $E_2$ page is too sparse for any differential, as the degree of a differential $d_r$ is $(t-s,s) = (-1,r)$. The $E_2$ page is an $\text{Ext}^{*,*}_{A(1)}(\mathbb{Z}_2, \mathbb{Z}_2)$ module by the so-called Yoneda product. The vertical and sloped lines represent the action by $h_0, h_1 \in \text{Ext}^{*,*}_{A(1)}(\mathbb{Z}_2, \mathbb{Z}_2)$ with degree $(t-s,s) = (0,1)$ and $(t-s,s) = (1,1)$ respectively. These actions are known to commute with differentials, so they can restrict possible differentials. Furthermore, the $h_0$ action remaining in the $E_\infty$ page indicates the extension.[32] Therefore, the "$h_0$ tower" starting from $(t-s,s) = (5,2)$ represents $\mathbb{Z}_2^\wedge$ (the module of 2-adic integers which is infinitely generated over $\mathbb{F}_2$, *not* the module with two elements). All in all, we obtained the following result:

$$\widetilde{\text{ko}}_d(SU(3)/SO(3))_2^\wedge = 0,0,\mathbb{Z}_2,\mathbb{Z}_2,0,\mathbb{Z}_2^\wedge,0,0, \quad \text{for } d = 0,1,2,3,4,5,6,7 . \qquad \text{(C.8)}$$

The fact that the $h_0$ tower representing the free part in $\text{ko}_5$ starts from $s = 2$ indicates that the generator of the free part is $2^2 = 4$ times the generator of the free part of the ordinary homology of the space.[33] Dually, this means that the WZW term corresponding to the generator $y_5 \in \mathbb{Z} \subset H^5(SU/SO; \mathbb{Z})$ should be divisible by 4, as noted more directly in Appendix D.1.

---

[32] There can be extensions which do not come from $h_0$, called *exotic* extensions. In our case the spectral sequence is too sparse for any exotic extension.

[33] One can get the spectral sequence converging to the 2-localized version of the ordinary homology $H(-;\mathbb{Z})$ by replacing $A(1)$ by $A(0) = \mathbb{F}_2[Sq^1]/((Sq^1)^2)$ in (C.1). This spectral sequence is equivalent to the (2-localized) Bockstein exact sequence. Now, in that spectral sequence, the $h_0$ tower starts from $(t-s,s) = (5,0)$, indicating the generator $y_5$ in the cohomology with $\mathbb{Z}$ coefficient is the $\mathbb{Z}$ uplift of $w_2 w_3$. From the naturality of the Adams spectral sequence, the map $\text{ko} \to H\mathbb{Z}$ is induced, in the way compatible with $h_0$, from the map between the corresponding map between the $E_2$ pages. Therefore, the image of the map $\text{ko} \to H\mathbb{Z}$ is generated by four times the dual of $y_5$. This argument does not determine the possible further multiplicity with prime factors other than 2.

$N_f \geq 7$ :   Now we consider the module $M' = \tilde{H}^*(SU/SO; \mathbb{Z}_2)$, which has new generators $w_4$ and $w_6$. The module would look like up to degree 7:

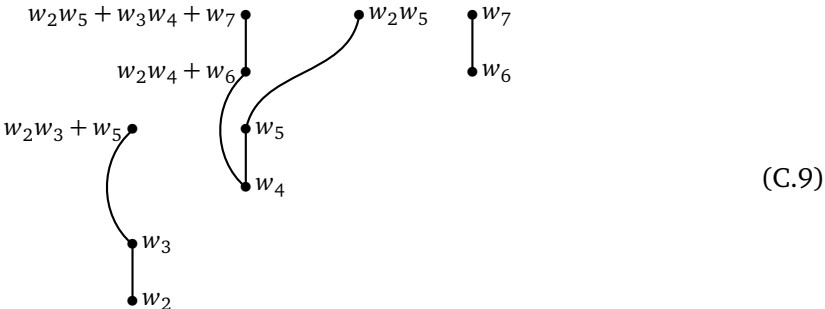

$$(C.9)$$

That is, we have $M'_{\leq 7} \cong (M \oplus A(1)[4] \oplus A(1)[6])_{\leq 7}$ as $A(1)$-modules, where $_{\leq 7}$ denotes the truncation at the degree 7. Therefore the minimal resolution of $M'$ looks like

$$0 \leftarrow M' \leftarrow A(1)[2] \oplus A(1)[4] \oplus A(1)[6] \oplus \text{higher degree} \leftarrow J[4] \oplus \text{higher degree} \leftarrow \cdots , \quad (C.10)$$

where the bottom element of $A(1)[4]$ is mapped to $w_4$ and that of $A(1)[6]$ is mapped to $w_6$. Therefore, the Adams chart is

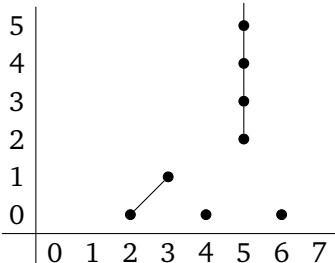

Although just from the degrees of the dots, there can be a non-trivial differential $d_2$ from $(t-s, s) = (6,0)$ to $(5,2)$, such a differential is not consistent with the $h_0$ action and hence is absent. The ko groups are

$$\widetilde{ko}_d(SU/SO)_2^\wedge = 0, 0, \mathbb{Z}_2, \mathbb{Z}_2, \mathbb{Z}_2, \mathbb{Z}_2^\wedge, \mathbb{Z}_2, 0, \quad \text{for } d = 0, 1, 2, 3, 4, 5, 6, 7 . \quad (C.11)$$

$N_f = 6$ :   By omitting $w_7$ from $M'$, one finds $H^*(SU(6)/SO(6); \mathbb{Z}_2)_{\leq 7} = (M \oplus A(1)[4] \oplus \mathbb{Z}_2[6])_{\leq 7}$ as $A(1)$ module. The Ext group for the trivial module $\mathbb{Z}_2$ can be found, e.g. in [78, Fig. 20]. Therefore its Adams chart is

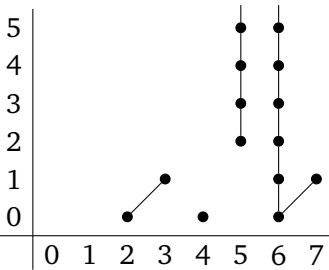

There cannot be any differential[34], and thus ko groups are

$$\widetilde{ko}_d(SU(6)/SO(6))_2^\wedge = 0, 0, \mathbb{Z}_2, \mathbb{Z}_2, \mathbb{Z}_2, \mathbb{Z}_2^\wedge, \mathbb{Z}_2^\wedge, \mathbb{Z}_2 \quad \text{for } d = 0, 1, 2, 3, 4, 5, 6, 7 . \quad (C.12)$$

---

[34]The $h_0$ tower at $t-s = 6$ cannot go into the tower at $t-s = 5$ because we see the free part at $d = 6$ from the AHSS.

$N_f = 5$ : By omitting $w_6$ and $w_7$ from $M'$, the result is the same as the $N_f \geq 7$ case, except that the degree 6 is now 0.

$N_f = 4$ : By omitting $w_5$, $w_6$ and $w_7$ from $M'$, one finds $H^*(SU(4)/SO(4); \mathbb{Z}_2) = M \oplus Q[4]$ as $A(1)$ module, where the module $Q$ ("question mark upside-down") is

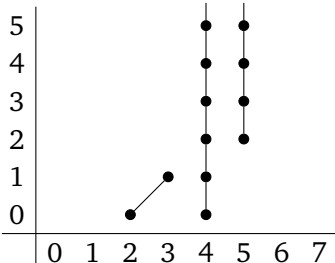

(C.13)

whose Adams chart is again found in [78, Fig. 29]. Therefore its Adams chart is

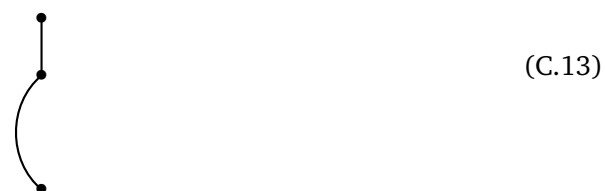

Although this $E_2$ page alone does not determine the differential from the $h_0$ tower at $t-s = 5$ to the tower at $t-s = 4$, it is prohibited because from AHSS we expect $\mathbb{Z} \subset \widetilde{\mathrm{ko}}_4(SU(4)/SO(4))$. Thus, the ko groups are

$$\widetilde{\mathrm{ko}}_d(SU(4)/SO(4))_2^\wedge = 0, 0, \mathbb{Z}_2, \mathbb{Z}_2, \mathbb{Z}_2^\wedge, \mathbb{Z}_2^\wedge, 0, 0, \quad \text{for } d = 0, 1, 2, 3, 4, 5, 6, 7 .$$ (C.14)

# D   More on the $SO$ WZW term

## D.1   Divisibility via $KO$-theory

Here we provide a derivation using $KO$-theory that the generator $y_5 \in \mathbb{Z} \subset H^5(SU/SO; \mathbb{Z})$ and for any map $f : W_5 \to SU/SO$ from a spin manifold $W_5$,

$$\int_{W_5} f^*(y_5) \in \mathbb{Z}$$ (D.1)

is divisible by 4. A different derivation using the Adams spectral sequence was given in Appendix C.

We first note that the quantity (D.1) is a $\mathbb{Z}$-valued spin bordism invariant. Therefore, any such $\mathbb{Z}$-valued function is determined by its value on the generator of the free part $\mathbb{Z}$ of $\Omega_5^{\mathrm{spin}}(SU/SO)$. To find it, use the fact that $SU/SO$ is basically $U/O$, which happens to be a classifying space of $KO^1$. Then, a map $f : W_5 \to SU/SO$ determines a class $[f] \in KO^1(W_5)$.

One also needs the integral in the sense of $K$-theory, not in the sense of ordinary (co)homology. In the case of ordinary integral cohomology groups, a cohomology class $u \in H^d(W_d; \mathbb{Z})$ can be integrated against the fundamental class of the manifold $[W_d] \in H_d(W_d; \mathbb{Z})$, which requires an orientation on the manifold. One then gets $\int_{W_d} u \in \mathbb{Z}$. These concepts generalize to $K$ and $KO$-theory. A $KO$-orientation of $W_d$ is a spin structure and defines the $KO$-theoretic fundamental

class $[W_d] \in KO_d(W_d)$. Then, a class $u \in KO^i(W_d)$ can be integrated to give

$$\int_{W_d} u \in KO^{i-d}(pt). \tag{D.2}$$

The major difference from ordinary (co)homology is that it can be non-zero even when $i \neq d$, since we have

$$\begin{array}{c|cccccccc} d \pmod 8 & 0 & -1 & -2 & -3 & -4 & -5 & -6 & -7 \\ \hline KO^d & \mathbb{Z} & \mathbb{Z}_2 & \mathbb{Z}_2 & 0 & \mathbb{Z} & 0 & 0 & 0 \end{array}. \tag{D.3}$$

These $KO$-theoretic integral reduces to the Atiyah-Singer index and mod 2 index, respectively.

Now one can consider

$$\int_{W_5} [f] \in KO^{-4}(pt) = \mathbb{Z}, \tag{D.4}$$

which is also a $\mathbb{Z}$-valued spin bordism invariant. Recall that $\widetilde{KO}^1(S^5) = \pi_5(U/O) = \mathbb{Z}$ and take its generator $f_0 : S^5 \to U/O$. From the Bott periodicity, this class integrates to one: $\int_{S_5} [f_0] = 1 \in \mathbb{Z} = KO^{-4}(pt)$. This means that $(S^5, f_0)$ is the generator of the $\mathbb{Z}$ part of $\Omega_5^{\mathrm{spin}}(SU/SO)$.

So one just have to check $\int_{S_5} f_0^*(y_5) = 4$ holds. One way to see this is to consider the map

$$U(\infty) \twoheadrightarrow U(\infty)/O(\infty) \hookrightarrow U(\infty), \tag{D.5}$$

sending

$$U \mapsto [U] \mapsto V = UU^{\mathsf{T}}. \tag{D.6}$$

Applying $\pi_5$ to (D.5), we obtain the sequence

$$\underbrace{\pi_5(U(\infty))}_{\mathbb{Z}} \longrightarrow \underbrace{\pi_5(U(\infty)/O(\infty))}_{\mathbb{Z}} \longrightarrow \underbrace{\pi_5(U(\infty))}_{\mathbb{Z}}, \tag{D.7}$$

where it is known [67] that it is an isomorphism at the first step, and a multiplication by two in the second step of the sequence. Applying $H^5(-;\mathbb{Z})$ to (D.5) instead, we get the sequence

$$\underbrace{H^5(U(\infty);\mathbb{Z})}_{\mathbb{Z}} \longleftarrow \underbrace{H^5(U(\infty)/O(\infty);\mathbb{Z})}_{\mathbb{Z} \oplus \mathbb{Z}_2} \longleftarrow \underbrace{H^5(U(\infty);\mathbb{Z})}_{\mathbb{Z}}, \tag{D.8}$$

where it is known [70] that the generator $x_5 \in H^5(U(\infty);\mathbb{Z})$ pulls back to the generator $y_5 \in H^5(U(\infty)/O(\infty);\mathbb{Z})$ which further pulls back to $2x_5 \in H^5(U(\infty);\mathbb{Z})$. Recalling that $x_5$ integrates to 2 on the generator of $\pi_5(U)$, one finds that $y_5$ integrates to 4 on the generator of $\pi_5(U/O)$.

Before ending, we note that the same argument can be applied to the divisibility of $x_5 \in H^5(U;\mathbb{Z})$ on a spin manifold. Indeed, $U(\infty)$ is the classifying space of $K^1$, and therefore one can compare $\int_{W_5} f^*(x_5) \in \mathbb{Z}$ and $\int_{W_5} [f] \in K^{-1}(pt) = \mathbb{Z}$. One again finds that the generator $\pi_5(U) = \mathbb{Z}$ is the generator of $\Omega_5^{\mathrm{spin}}(U)$. Then one only needs to evaluate $\int_{M_5} f_0^*(x_5)$ on this particular case.

## D.2 Fixing the torsion part

In Sec. 2.6 and in Sec. 2.8.5, we identified the $SO$ WZW terms using the sequence

$$0 \to \underbrace{\mathrm{Ext}_{\mathbb{Z}}(\Omega_4^{\mathrm{spin}}(X), \mathbb{Z})}_{=\mathbb{Z}_2 \text{ or } 0} \to (D\Omega_{\mathrm{spin}})^5(X) \to \underbrace{\mathrm{Hom}_{\mathbb{Z}}(\Omega_5^{\mathrm{spin}}(X), \mathbb{Z})}_{=\mathbb{Z}} \to 0. \tag{D.9}$$

The analysis in Sec. 4.2 then showed that the WZW term for the $SO(N_c)$ QCD corresponds to $N_c$ times the generator of the free part $\mathbb{Z}$ in the above sequence. But we have not succeeded in fixing the torsion part. In principle, the path integral of $Spin(N_c)$ QCD determines the invertible phase including the torsion part, but this is highly non-perturbative and does not provide any effective way to compute it.

Here we discuss a tentative direction.[35] We start by noting that the $Spin(N_c)$ QCD contains fermions $\psi_{ai}$ where $a = 1,\ldots,N_c$ and $i = 1,\ldots,N_f$. The sigma model field arises as $\Lambda^3 \sigma_{ij} = \langle \psi_{ai}\psi_{aj} \rangle$ where $\sigma_{ij}$ is a complex symmetric matrix and $\Lambda$ is the dynamical scale. The WZW term is what reproduces the $SU(N_f)$ anomaly in the low-energy sigma model.

Now let us introduce $N_c N_f$ spectator fermions $\tilde{\psi}_s^i$ where $i = 1,\ldots,N_f$ is in the anti-fundamental representation of the flavor symmetry and $s = 1,\ldots,N_c$ is in the vector representation of an $SO(N_c)$ symmetry *distinct* from the gauge symmetry. We also introduce the four-fermion term $\psi_{ai}\psi_{aj}\tilde{\psi}^{si}\tilde{\psi}^{sj}$. In the low-energy limit, this becomes a mass term $\Lambda^3 \sigma_{ij}\tilde{\psi}^{si}\tilde{\psi}^{sj}$ for the spectator fermions. Then the low-energy limit is gapped with a unique vacuum.

Since the $Spin(N_c)$ QCD plus the spectator fermions do not have any $SU(N_f)$ anomaly, we expect that the total system in the infrared is a trivial invertible phase. Then the $SO$ WZW term should simply be (the inverse of) the phase of $N_c$ fermions with a position-dependent mass specified by the sigma model field $\sigma_{ij}$. Then we propose the following identification:

$$e^{-N_c[\sigma:M_4 \to SU(N_f)/SO(N_f)]} = \left( \lim_{m\to+\infty} \frac{\det \slashed{D} + m\delta_{ij}}{\det \slashed{D} + m\sigma_{ij}} \right)^{N_c}. \tag{D.10}$$

The right hand side is manifestly well-defined once a spin structure is given, and clearly has the correct anomalous variation with respect to $SU(N_f)$ flavor symmetry.

This should also automatically imply that $\int_{M_5} \sigma^*(y_5)$ is a multiple of four, because otherwise the WZW term would be inconsistent. Arguably, this statement should also follow from the consistency of the path integral of QCD and from its low-energy representation by the sigma model, but it is too strongly-coupled to make into a mathematical statement at present. The right hand side of (D.10) is still a result of a path integral but of a free theory, and is far easier to analyze than the strongly-coupled QCD. It should be possible to make this mathematically precise. The right-hand side of (D.10) is a generalization of the exponentiated eta invariant to the case with position-dependent mass term. Such mathematical objects seem useful in generating spin invertible phases and deserve a more detailed study.

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
