# Peer review of "Revisiting Wess-Zumino-Witten terms"

_SciPost Physics, doi:SciPost Phys. 10, 061 (2021)_

## Round 2 · Referee Report · Dileep Jatkar · 2020-12-28

Strengths

Comprehensive study of the Wess-Zumino terms both in the gauged and non-gauged cases.

Weaknesses

Nothing very particular.

Report

In this manuscript authors do a comprehensive study of non-gauged as well as gauged Wess-Zumino terms in four dimensions. It starts with a warm up exercise in two dimensions which sets the stage for the four dimensional analysis. This helps put the 4D results in perspective in an otherwise technical manuscript. The results are relevant for study of topological phases both in condensed matter physics as well as in the low energy description of gauge theories in terms of sigma models. The manuscript yo-yos between mathematical details and physical insight, which is a good format for both a mathematically minded reader as well as for a physicist.

Overall the manuscript meets the SciPost publication criterion and I recommend publication of the manuscript.

Requested changes

Changes are minor.

There are a few places the statements are a bit confusing. For example, on page 11, the sentence referring to the Atiyah-Hirzebruch spectral sequence (reproduced below) is a bit confusing.

"That this construction detects Ωspin(SU(2)) = Z can be seen
by studying the Atiyah-Hirzebruch spectral sequence (AHSS) computing it."

---

## Round 2 · Referee Report · Anonymous · 2021-2-7

Report

A four dimensional QCD theory where the flavor symmetry is spontaneously broken, is described by a nonlinear sigma model on the coset space. The non-zero flavor anomaly in the UV is matched by the so-called Wess-Zumino-Witten (WZW) terms in the low energy sigma model. These terms have been studied to a great extent in two dimensions, in this paper, authors perform a systematic study of these terms in four dimensions. In particular they show that in order to define WZW terms, even without background gauge fields, the manifold needs to have a spin structure. A four dimensional Spin QCD also has a one-form global symmetry which is responsible for stability of electric flux tube operators. The authors study the interplay of the WZW term with this symmetry. In particular, they describe how the mixed anomaly is reproduced in the low energy sigma model description. In addition to these two main points, the paper reviews all the background necessary which makes their treatment of anomalies self-contained. 
First the authors discuss WZW terms in two dimensions. The target space of the sigma model is a group G and the WZW terms are written as the integral of the pullback of the normalized G-invariant 3-form over D3 where D3 is some 3-manifold that is bounded by the string worldsheet. Following the same strategy, the WZW term in four dimensions is defined as the integral of the pullback of the normalized G-invariant 5-form over W5 where W5 is some 5 manifold that is bounded by the four manifold M4 on which the theory lives. In this case, the authors argue that the canonical five form is normalized with respect to homotopy but not homology. That is why it integrates to 1/2 rather than 1 on closed five manifolds. They confirm this by considering a canonical map from the Wu 5-manifold = SU(3)/SO(3) into SU(3). This means that the WZW term, defined above, has a sign ambiguity. Authors show that if the manifold has a spin structure this sign ambiguity is absent. They do this in two steps. First they show that if closed W5 is spin then the integral of 5-form is integer valued (rather than half-integer). Then, using bordism theory, they show that if M4 is spin then extension of the map from W5 exists where W5 is also spin. The case of G=SU(2) requires separate treatment because dim(SU(2)) is 3 and it does have the G-invariant 5-form used above.
After discussing anomalies in QCD and gauged WZW terms, the authors discuss the mixed anomaly between 0-form baryonic symmetry and the flavor symmetry for SU QCD and the mixed anomaly between Z_2 1-form symmetry and the flavor symmetry for Spin QCD. They explain how these anomalies are reproduced in the low energy sigma model. The paper is supplemented by four appendices reviewing computation of cohomology, bordism groups of classifying spaces and homogeneous spaces among other relevant topics.
The paper is well-written. In addition to making significant contributions towards understanding WZW terms in higher dimensions, the paper also reviews existing literature on WZW terms in modern language. We recommend this paper for publication.

---

## Editorial Decision

published